# Relative sky radiance from multi-exposure all-sky camera images

Juan C. Antuña-Sánchez[1], Roberto Román[1], Victoria E. Cachorro[1], Carlos Toledano[1], César López[2], Ramiro González[1], David Mateos[1], Abel Calle[1], Ángel M. de Frutos[1]

[1]Group of Atmospheric Optics, Universidad de Valladolid (GOA-UVa), Valladolid, 47011, Spain
[2]Sieltec Canarias S.L., La Laguna, 38230, Spain

*Correspondence to*: Juan Carlos Antuña Sánchez (jcantuna@goa.uva.es)

**Abstract.** All-sky cameras are frequently used to detect cloud cover; however, this work explores the use of these instruments for the more complex purpose of extracting relative sky radiances. An all-sky camera (SONA202-NF model) with three colour filters, narrower than usual for this kind of cameras, is configured to capture raw images at seven exposure times. A detailed camera characterization of the black level, readout noise, hot pixels and linear response is carried out. A methodology is proposed to obtain a linear high dynamic range (HDR) image and its uncertainty, which represents the relative sky radiance (in arbitrary units) maps at three effective wavelengths. The relative sky radiances are extracted from these maps and normalized dividing every radiance of one channel by the sum of all radiances at this channel. Then, the normalized radiances are compared with the sky radiance measured at different sky points by a sun/sky photometer belonging to the Aerosol Robotic Network (AERONET). The camera radiances correlate with photometer ones except for scattering angles below 10º, which is probably due to some light reflections on the fisheye lens and camera dome. Camera and photometer wavelengths are not coincident, hence camera radiances are also compared with sky radiances simulated by a radiative transfer model at the same camera effective wavelengths. This comparison reveals an uncertainty on the normalized camera radiances about 3.3%, 4.3% and 5.3% for 467, 536 and 605 nm, respectively, if specific quality criteria are applied.

# 1 Introduction

The knowledge of sky radiance is a fundamental problem of the radiative transfer in the atmosphere, or other media, where absorption, emission and scattering processes occur (Coulson, 1988). Restricting to the case of solar radiation in the atmosphere-surface of the Earth, sky radiance depends on the Sun position in the sky and its angular distribution is mainly controlled by the light scattering caused by atmospheric gases through the Rayleigh scattering (responsible for the colour of the blue sky under clear conditions) but also caused by aerosols and clouds through Mie scattering. The knowledge of the sky radiance is useful, among other fields, in photovoltaic production, to calculate what solar radiation reaches an oriented panel (Li and Lam, 2007) and in human health to know the solar UV radiation dose received by a human body (Seckmeyer et al., 2013; Schrempf et al., 2017).

The spectral sky radiance reaching the Earth surface under cloud-free conditions is basically the solar irradiance scattered by gases and aerosols, therefore the knowledge of the spectral sky radiance at different angles is a footprint of the aerosol properties; it implies the sky radiance contains useful information that can be used for the retrieval of aerosol optical and microphysical properties (Nakajima et al., 1996; Dubovik and King, 2000). In fact, even relative sky radiance measurements (in arbitrary units) are useful for this purpose (Román et al, 2017a). Most of remote sensing techniques, mainly those used by satellite platforms, are also based on upward sky radiance measurements, formed by the radiation reflected by the earth surface and scattered by the atmosphere, allowing to determine the different atmospheric compounds.

Accurate measurements of the sky radiance are usually taken by photometers. As an example, the CE318 sun/sky photometer (*Cimel Electronique S.A.S.*), which is the reference instrument of the Aerosol Robotic Network (AERONET; https://aeronet.gsfc.nasa.gov), measures the absolute sky radiance at different geometries and wavelengths with an uncertainty about 5% (Holben et al., 1998). These sky radiance scans provide useful and accurate information (e.g., AERONET use them to retrieve and provide aerosol products); however, the recorded sky radiances are only measured at some sky angular positions and the more points are measured, the more time is spent; it causes a temporal shift between measurements and the sky scene can change during this time.

All-sky cameras are devices designed to capture images of the full hemispherical sky, consisting usually of a CCD or CMOS sensor looking to a mirror or with a mounted fisheye lens. The most frequent use of all-sky cameras is the cloud cover detection (e.g., Tapakis and Charalambides, 2013 and references therein), but they have also been used for more complex purposes like solar irradiance forecasting (Alonso-Montesinos et al., 2015; Barbieri et al., 2017); to derive sky radiance and luminance measurements (Román et al., 2012; Tohsing et al., 2013); to retrieve aerosol properties (Cazorla et al., 2008; Román et al., 2017a); and to monitor aurora and airglow (Sigernes et al., 2014), among others. All-sky cameras are in general less accurate than well calibrated photometers, but they are capable to obtain a full map of the hemispherical sky radiance in a short time (few seconds or less). In addition, the camera sensors allow the variation of exposure time and gain, achieving a high dynamic range. These facts and the mentioned versatility of the all-sky cameras lead to consider these devices as a complementary

instrument of sun/sky photometers and as a cheaper alternative to perform sky radiance measurements in locations where photometers are not available.

This framework motivates the main objectives of this paper: to characterize the main properties of an all-sky camera and to develop a methodology to obtain relative sky radiances from this camera. This work also aims at quantifying the uncertainty of the sky radiances obtained by the proposed methodology through a direct comparison with photometer measurements as well as with simulated radiances. The novelty of this work with respect to other works that also retrieve the sky radiance with sky cameras, such as Román et al. (2012), lies among other issues in the use of raw and multi-exposure images captured with narrower spectral filters.

This paper is structured as follows: Section 2 introduces the experimental site and the instrumentation. Section 3 describes in detail the methodology that has been developed to retrieve the relative sky radiances with the camera, while Section 4 presents the comparison of camera radiances with photometer measurements and simulated radiances. Finally, the main conclusions are summarized in Section 5.

## 2 Site and instrumentation

All the measurements used in this work were carried out in a platform located on the rooftop of the Science Faculty of Valladolid, Spain; (41.6636ºN, 4.7058ºW; 705 m a.s.l.). Valladolid, sited in North-Central Iberian Peninsula (150 km North from Madrid), is an urban city with a population around 300,000 inhabitants (~400,000 including the metropolitan area). The city is surrounded by rural areas and it shows a Mediterranean climate (Csb Köppen classification) with hot summers and cold winters. The predominant aerosol at Valladolid is classified as "clean continental" (Bennouna et al., 2013, Román et al., 2014) but occasionally Saharan dust particles are transported to Valladolid, especially in summer (Cachorro et al., 2016).

The instrumentation at the mentioned platform is managed by the "Grupo de Óptica Atmosférica" (Group of Atmospheric Optics) of the University of Valladolid (GOA-UVa). The GOA-UVa is in charge of the calibration of part of the AERONET photometers, hence two reference photometers -accurately calibrated by the Langley-plot at the high-altitude station Izaña (Toledano et al., 2018)- and various field photometers under calibration, are always operative at the platform. All the photometers are different versions of the CE318 (*Cimel Electronique S.A.S.*). The most recent model is the CE318-T sun/sky/moon photometer, which allows measurements of direct solar and lunar irradiance to derive the aerosol optical depth (AOD) at several wavelengths. This photometer (and older versions) is also capable of taking measurements of sky radiance at various wavelengths. The CE318-T mainly makes sky radiance scans at two different configurations: almucantar (zenith angle equal to solar zenith angle, SZA, while azimuth angle varies) and hybrid (a mix between almucantar and principal plane) scans (Sinyuk et al., 2020). Both configurations present spatial symmetry regarding the Sun position, which is useful to reject cloud contaminated measurements comparing left and right observations. CE318-T usually is configured to measure sky radiances at 380, 440, 500, 675, 870, 1020 and 1640 nm thanks to narrow interference filters mounted in a filter wheel. The multi-wavelength sky radiance measurements are used by AERONET to retrieve aerosol properties by inversion techniques

(Dubovik et al., 2000; Sinyuk et al., 2020), like aerosol size distribution, complex refractive indices, and the fraction of spherical particles (sphericity factor).

This work uses the photometer sky radiances at 440, 500 and 675 nm measured at Valladolid for both AERONET almucantar (only for SZA>40º) and hybrid scenarios. These data have been directly obtained from AERONET website (Aerosol inversions v3 - Download Tool), level 1.5 version 3 data. The size distribution, refractive indices, and sphericity factor products of the version 3 AERONET level 1.5 (Sinyuk et al., 2020) have been also downloaded for this work.

The mentioned GOA-UVa platform is also equipped with a SONA202-NF (*Sieltec Canarias S.L.*) all-sky camera. This device is a prototype that mainly consists of a CMOS coupled to a fisheye lens, both encapsulated in a weatherproof case with a transparent glass dome. The camera is horizontally levelled to receive the sky radiance of the full hemispherical sky. The CMOS sensor is the SONY IMX249 and is configured to save raw images of 1172x1158 pixels with a resolution of 10 bits. This sensor has a Bayer filter mosaic following a RGGB pattern: half of the pixels are mainly sensitive to Green (G), a fourth of them to Red (R), and the other fourth to Blue (B). The spectral response of these filters, obtained from the data sheet from the CMOS manufacturer, is shown in Fig. 1a. An additional RGB triband filter (Fig. 1b; spectral response provided by the manufacturer) is over the full mosaic in the SONA202-NF in order to reduce the width of the colour filters. As result Fig. 1c shows the final spectral response of the Red, Green, and Blue pixels which is narrower than without the triband filter; this additional filter also reduces (but not fully eliminates) the overlapping between the colour channels.

The sensor allows taking pictures at different time exposures and signal amplifications (ISO). This all-sky camera is configured to take, every 5 minutes, a set of raw images with different exposure times in order to have enough pixel signal without saturation in the brightest sky parts (lower exposure times) and in the darkest (higher exposure times). Two different exposure configurations are set in the camera: daytime and night-time, due to the need of different exposure times, but this paper is only focused on daytime mode which was assumed for all images with SZA below 95º. No amplification is used in daytime mode. The exposure times used for each set of multi-exposure raw images are: $t_1=0.3\mu s$; $t_2=0.4\mu s$; $t_3=0.6\mu s$; $t_4=1.2\mu s$; $t_5=2.4\mu s$; $t_6=4.8\mu s$; and $t_7=9.6\mu s$. These exact exposure values are entered in the camera software, however, some tests varying these values indicated that the real exposure times could be discretized, showing equal images for different but close exposure times. Conversely, other images showed significant jumps in the recorded signal from images with small differences in the introduced exposure times.

The time expended to take the seven daytime raw images is few seconds and, after they are recorded, all images are saved together with additional metadata (such as sensor temperature) in one *.h5 file to reduce memory. Finally, the camera has been geometrically calibrated using a set of cloud-free night-time images in the ORION software (Antuña-Sánchez et al., 2020), which determines the position of the sky (azimuth and zenith) viewed by each pixel, and its field of view (FOV), through the star positions in the images.

## 3 Method

### 3.1 Effective wavelengths

The three camera channels are sensitive to a broadband range of wavelengths because of the width of their spectral response (as discussed above, see Fig. 1). However, the measured broadband radiance can be assigned to an effective wavelength assuming the recorded broadband signal is proportional to the radiance at this effective wavelength (Román et al., 2017a). The ratio of two broadband measurements, which are taken under different conditions but with the same instrument (the same spectral response), is equal to the ratio of the same measurements taken with an instrument which is only sensitive at the effective wavelength (Kholopov, 1975). The effective wavelength of each channel can be calculated by the convolution of the measured radiance by the channel spectral response, as was explained by Román et al. (2017a).

The effective wavelength of each channel has been calculated in this work for 200 different sky scenarios following the same method as in Román et al. (2012). To this end, the spectral diffuse sky radiance has been simulated using libradtran 1.7 radiative transfer package (Mayer and Kylling, 2005) for: SZA values from 10º to 80º (in 10º steps), Angström Exponent values from 0.2 to 1.8 (in 0.4 steps), and turbidity coefficient values (AOD at 1000 nm) from 0.01 to 0.21 (in 0.05 steps). Each simulation has been used to obtain an effective wavelength at each channel, given a total of 200 different effective wavelengths per channel. The median (±standard deviation) of all obtained effective wavelengths is 605±3 nm, 536±3 nm and 467±2 nm for the Red, Green, and Blue channels, respectively. These obtained values have been assumed as the effective wavelengths of the analysed camera. These results are similar to those obtained by Román et al. (2012), especially at G and B channels, for other all-sky camera model with a wider spectral response.

### 3.2 White balance

The channel spectral response affects the final colour image after a demosaicing (interpolation of the pixel signals of a channel to the pixels of the other channels) of the raw recorded image. Hence, a white balance is frequently used to obtain a final true colour demosaiced image. White balance mainly consists of multiplication of the recorded signal at each channel by a scaling factor (WBSF) which is different for each R, G and B channel; it weights the relationship between the three colour channels achieving a realistic final colour. The SONA202-NF used in this work was configured to provide a raw image with a previous white balance applied, where the G and B channels are multiplied by ~1.1 and 2.1, respectively, while R channel remains the same (multiplied by 1).

This fact makes that the direct demosaiced image of the analysed camera looks more realistic. It can be observed in the example pictures of Fig. 2, where Fig. 2a shows a direct image with applied white balance showing a more realistic blue sky colour than Fig. 2b, where the same picture is shown but with G and B channels divided by their white balance scaling factors. This image presents a more greenish and less bluish sky. A true colour image is useful for the all-sky camera primary objective of detecting clouds. However, the application of the white balance scaling factors before obtaining the raw image reduces the

image dynamic range since it can saturate pixel signals which initially were not saturated, especially at the Blue channel in this case, because the signal is multiplied by the mentioned factor of 2.1.

### 3.3 Dark signal and hot pixels

The recorded signal of each pixel depends on the received light (photons converted to electrons), but some pixel signal appears in the recorded images even in the absence of light. This signal without light is the called readout noise and it is caused by camera electronics (amplification, analog to digital conversion, etc.) in the readout process. The readout noise is gaussian distributed, hence the cameras usually add an offset (black level) to the recorded raw values in order to detect also negative readout noise values (signal below the offset). Summarizing, the recorded signal in one pixel is the signal generated by the

number of absorbed photons plus the black level signal (offset) plus the total noise (being the readout noise part of the total noise).

The analysed camera was covered for two and half days with a metal piece designed to block all the incoming sky light. Meanwhile the camera was capturing multi-exposure images as in its operational routine. A total of 413 images (dark frames) per exposure time were acquired in daytime mode. The Red channel of all these dark frames has been used to determine the

black level since this channel is the only one not affected by the mentioned white balance process. The mode and median of all red pixels are 30 digital counts (DC) for each one of the 2891 (413 dark frames x 7 exposures) measured dark frames at the different time exposures; this reveals the black level of the sensor is 30 DC. Hence, the signal at the R, G and B channels has been corrected for black level offset and white balance by the next equation:

$$R_c = \frac{R_{raw} - 30}{WBSF_R}; G_c = \frac{G_{raw} - 30}{WBSF_G}; B_c = \frac{B_{raw} - 30}{WBSF_B}; (1)$$

where $R_c$, $G_c$, and $B_c$ are the black level and white balance corrected signal of R, G and B channels, respectively; $R_{raw}$, $G_{raw}$, and $B_{raw}$ are the recorded raw camera signals while $WBSF_R$, $WBSF_G$ and $WBSF_B$ are the white balance scaling factors of the R ($WBSF_R$=1), G ($WBSF_G$=1.1) and B ($WBSF_B$=2.1) channels, respectively. After this correction, the readout noise must be the same for the three channels.

All the dark frames have been corrected by Eq. (1), and the mean ($M_{DFS}$; mean dark frame signal) and standard deviation ($\sigma_{DFS}$) of the signal of all pixels (including the three channels) has been calculated for each dark frame. Fig. 3 (upper panels) shows these values as a function of temperature and for the different exposure times. As can be observed there is no dependence on the exposure time for the mean neither for the standard deviation, which represent the readout noise of each dark frame. On the other hand, the mean and standard deviation increase with temperature, but the mean values show low values around 0.10-

0.13 DC. Moreover, the mean values slightly decrease for temperatures above 50ºC while the standard deviation still increases up to values around 0.65 DC for 55ºC.

These results do not provide information about the spatial distribution of the pixel signal in a dark frame. Hence, an averaged dark frame (ADF), in which each pixel is the mean of this pixel signal in each recorded dark frame, is shown in Fig. 4a. Vertical column patterns are observed, as was also described by Román et al. (2017a), but in general with low values and without a significant signal variation that is likely linked to the readout process of the camera. However, some pixels with much higher signal appear in this picture; these are known as "hot pixels".

The hot pixels present a high signal even without light, and this signal usually increases with increasing temperature and exposure time (Porter et al., 2008). This dependence on temperature has been used to identify and reject the hot pixels of the camera. The correlation coefficient (r) of each pixel with the temperature at each exposure time for all available dark frames has been calculated. Figure 5a shows the correlation coefficient for the $t_1$ exposure time. The spatial distribution of this correlation coefficient is similar to the ADF, showing some column patterns but with most of the values near zero, and with high values for hot pixels. In fact, the frequency distribution of this correlation coefficient, also shown for $t_1$ in Fig. 5b, presents a Gaussian distribution centred around zero, thus indicating no correlation with temperature. However, some pixels show high r values. A threshold value to detect those outliers has been defined as the median, multiplied by 2, minus the minimum of the r distribution. The pixels showing a 'r' value higher than this threshold in any of the seven exposure times have been classified as hot pixels and they will be excluded in the analysis of this work. A total of 2158 hot pixels (0.16% of the total) have been detected by this method. Fig. 4b shows the ADF without the identified hot pixels, and the reduction of the hot pixels is significant, indicating a good performance of the used method to detect hot pixels. Dead pixels, whose signal does not vary even under light presence, have not been found in the camera.

The $M_{DFS}$ and $\sigma_{DFS}$ values have been recalculated rejecting the identified hot pixels, and these values are also shown in Fig. 3 (bottom panels) as function of temperature. The $M_{DFS}$ values are similar with and without hot pixels, however, the $\sigma_{DFS}$ values, associated with the readout noise, are significantly reduced when the hot pixels are discarded. Without hot pixels, $\sigma_{DFS}$ shows a dependence on temperature similar to $M_{DFS}$, reaching a maximum value around 0.43DC close to 50ºC. Considering this result, the readout noise ($N_r$) of the analysed camera has been assumed as 0.43 DC.

Finally, for all pixel signals (excluding hot pixels) of all measured dark frames at all exposure times, 81.55% (20.44%, 40.64% and 20.47% for R, G and B) of the signals have 0 DC, 14.94% (3.69%, 7.56% and 3.69% for R, G and B) have 1 DC, and 3.51% (0.86%, 1.81% and 0.84% for R, G and B) have -1 DC. These results indicate the low readout noise of the camera, being most of the signals equal to zero and not showing a dependence on channel, which is expected if the black level and white balance are well corrected.

### 3.4 Linear response and effective exposure times

One of the most important sensor characterizations is the linear response of the pixel signal, which mainly depends on the structure of the pixel type (Wang, 2018). This feature indicates how linear the ratio between the pixel signal and the incoming irradiation is. The linear response can be obtained by varying the intensity of a light source with a fixed exposure time for the image sensor, or by increasing the exposure time (it increases the received irradiation) of the sensor at a fixed light condition

(Wang, 2018). The analysed camera is installed outdoors; hence the variation of exposure times under a fixed light conditions, such as sky light during few seconds, is more feasible than using a controllable and variable light source.

Debevec and Malik (1997) represented the pixel signal as function of the exposure time to retrieve the pixel linear response, finding nonlinearity at low and high pixel signal values. In our case, we have observed that the exposure times are discretized, hence we do not know the applied exposure times with accuracy. Therefore, we have represented each corrected pixel signal

($PS_c$) obtained at a given exposure time as a function of the same signal but captured under other exposure time (same pixel and scenario, but different exposure times and hence different signals); it has been done for all combinations of two different exposure times. Pixels viewing buildings or below the horizon have been masked and are not considered. Figure 6 shows these representations for the different exposure times using the pixels of all images recorded on 18th August 2019. The relationship between pixel signals at different exposure times looks linear. Pixels with (uncorrected) raw signal above 984 DC have been

assumed as saturated and they have been removed for all images and do not appear in Fig. 6. The 984 DC threshold is based on the visual inspection of graphs similar to Fig. 6 but without saturated pixel rejection. Higher data dispersion in Fig. 6 is observed when the exposure time difference is greater among them. This data dispersion is mainly caused by the presence of moving clouds on 18th August 2019, that can quickly vary the incoming radiation to the pixels. This is confirmed by Fig. A1, which is similar to Fig. 6 but for a cloud-free day (17th August 2019) and where the dispersion is lower. In spite the clouds,

the number of dispersed data in Fig. 6 is too low considering the high number of total data (18-112 million depending on the panels) and the log-scale of the number of data in the density plots.

The data of the Fig. 6 have been fitted to a linear regression for each panel using a weighted least square fit. The chosen weight, w, of each data pair has been:

$$w = \frac{1}{\sqrt{N_x{}^2 + N_y{}^2}} \qquad (2)$$

where $N_x$ and $N_y$ represents the total noise (N) of the $PS_c$ measurement for the exposure time represented in x and y axis, respectively. N can be described as the sum of readout noise and the shot noise ($N_s$). $N_s$ is associated with the particle nature of light and can be expressed as the square root of the measured signal caused by light (black corrected) since it follows a Poisson distribution. Hence, in this work:

$$N = \sqrt{N_r{}^2 + N_s{}^2} = \sqrt{0.43^2 + PS_c} \qquad (3)$$

The used linear fit has been chosen to weight the residual differences of a least square fit according the noise of the $PS_c$ data pairs. The linear fits of Fig.6 agree well with the data showing a high correlation with r values close to 1. The y-intercept of

250 all fits is close to zero, as expected, and it points out the goodness of the black level correction. The expected slope of each fit

should be the ratio between both exposure times because the recorded signal must be proportional to the integration time; however, the obtained slopes differ from the expected values. This result indicates that surely the nominal values of the used exposure times are not equal to the real exposure times of the sensor, as we suspected due to the observed discretization.

These slope and y-intercept values have been calculated for all available days between 12 July 2018 and $1^{st}$ May 2020 and only for the consecutive time exposures (cases shown in the diagonal plots of Fig. 6) because they show the lower deviation. A total of 56 days has been discarded because they show a correlation coefficient below 0.999 in at least one of the exposure times. Figure 7 shows the obtained remaining values for all period (a total of 593 data for exposure time relationship). The slope values do not present a significant variation in time, neither the y-intercept, being the most values below 1 DC. The number of data used for the shorter exposure times clearly vary with the sunshine duration, while for the higher exposure times the number of data are always similar, which is explained by the frequent pixel saturation reached for these higher exposure times, especially as the SZA decreases.

The slope and y-intercept correlation with the mean temperature has been also analysed, obtaining a r value ranging from -0.04 ($t_1$, $t_2$) to -0.69 ($t_5$, $t_6$) for the slope, and from 0.15 ($t_6$, $t_7$) to 0.64 ($t_3$, $t_4$) for the y-intercept. Despite the correlation between the slopes and temperature are not negligible, the standard deviation of the slopes is about 0.15% for all exposure times, which indicates a low variation. The empirical relationship between two exposure times has been assumed as the mean of the obtained slopes for these two exposure times, being the assigned uncertainty the combination of the standard deviation of the 519 obtained daily values and the propagated error in the slope values associated to the least square fits. The obtained exposure time relationships, given by the mean of the calculated slopes, provide a set of relative exposure times which achieve an effective linear pixel response; hence these relative exposure times can be assumed as effective exposure times for linear response. The mean slope obtained for the i and j exposure times is therefore equal to the ratio between these times: $t_j/t_i$.

**3.5 High dynamic range linear image**

The multi-exposure configuration was chosen to capture the maximum non-saturated signal of all-sky points from the brightest to the darkest, with the aim to form a unique image with high dynamic range (HDR), where the signal of each pixel will be linearly proportional to the received sky radiation. It means that in this image, the signal ratio between two pixels should be equal to the ratio of the sky radiation incoming to both pixels.

To this end, in this work one only linear HDR image has been calculated for each available image set formed by 7 multi-exposure raw images. The signal of a pixel of the HDR image is the $PS_c$ of the same pixel in the image where this pixel show the highest signal (lowest noise) but without saturation (original signal below 985 DC); this signal is then normalized to the $t_3$ exposure time. As example, if a pixel reaches the highest $PS_c$ without saturation at the image with $t_5$, then the signal assigned to the this pixel in HDR image will be the $PS_c$ in the $t_5$ image divided by $t_5/t_4$ and by $t_4/t_3$, both values obtained in Section 3.4 (the mean value of the slopes). If the highest $PS_c$ without saturation will be reached at the image with $t_1$ then the $PS_c$ in the $t_1$ image will be multiplied by $t_2/t_1$ and by $t_3/t_2$. The normalization to $t_3$ instead of other times has been chosen because the most of non-saturated pixels appear for exposure times between $t_1$ and $t_5$ (see Fig. 7c), hence $t_3$ reduces the number of multiplications

between coefficients, and hence the uncertainty. Usually one (from $t_2$ or $t_4$ to $t_3$) or two (from $t_1$ or $t_5$ to $t_3$) coefficient

multiplications are needed. The HDR signal of the pixels showing saturation in all image set is assumed as null value. The uncertainty on the HDR signal is calculated as the propagation of the uncertainties of $PS_c$ and of the applied coefficients to $t_3$ normalization.

    Figure 8 shows an example of the calculated HDR signal and its propagated uncertainty for each channel at Valladolid on 17th August 2018 07:25 UTC (same case than Fig. 2). Sun appears saturated in the three channels due to the high value of solar

radiation even in the lowest exposure time. The Blue channel presents higher values (regarding the maximum values) in the sky than the other channels; it is expected due to the bluish sky colour. The differences in the arbitrary unit scale between the three channels are caused by the white balance correction, which makes the Red channels to reach higher values than Green and Blue. The higher uncertainty in the Blue channel (mostly between 4.5 and 6.5%) is caused by the same reason. The uncertainty in the Red and Green channels ranges from 3.5% to 4.5%, but it shows a circular pattern in the middle of the image,

especially for the Red channel, which is not clearly appreciated in the Blue channel. This could be related with the formation of a reflected image of part of the camera in the dome, as can be seen in Fig. 2.

    The retrieved HDR images are 2D, with only one colour assigned to each pixel. This is enough to extract relative radiances, but to visualize a colour image of the captured scene can be of help for other applications. To this end, an RGB colour HDR has been retrieved from the HDR by a demosaicing algorithm (Li et al., 2008) which converts the Bayer pattern encoded image

into a true-colour image. The signal of this colour HDR image is still linear; hence its direct representation could only show the brightest areas of the sky. Therefore, a tone mapping (Salih et al., 2012) has been applied to this colour HDR to include all the dynamic range in the scale. As result, Fig. 2c shows the tone map of a colour HDR image. This image shows the solar aureole and the sky without saturation and with enough brightness. However, it looks greenish than the real sky. It is because in the HDR the white balance is not applied (it has been removed in the $PS_c$). To solve that, the Blue and Green channels of

the colour HDR image have been multiplied by $WBSF_B$ (1.1) and $WBSF_G$ (2.1), respectively, to apply the original white balance. The result is shown in Fig. 2d, showing a more realistic blue sky. This result indicates that the white balance can be applied after the raw image capture instead of before, giving a similar result but avoiding the saturation of several pixels and not reducing the dynamic range of the channels.

## 3.6 Extraction of relative sky radiance

Once a linear HDR image has been calculated, the relative sky radiance at any sky point can be extracted. The term "relative radiance" in this work refers to uncalibrated sky radiance measurements, i.e., it is the sky radiance but in arbitrary units instead of $Wm^{-2}sr^{-1}$ or physically equivalent units. First, all the HDR pixel signals are divided by their field of view to obtain radiance units, signal per solid angle ($sr^{-1}$). Then, for a sky point, defined by its zenith and azimuth coordinates, the great-circle distance between its coordinates and the coordinates viewed by each pixel is computed, and the camera pixel showing the lowest great-

circle distance is assumed as the pixel pointing to this sky point. This obtained pixel is only representative of one channel (R, G or B) and its signal could be noisy, hence a disk with radius of 3 pixels centred around the obtained pixel (a total of 37

pixels) is chosen to include the three channels. The chosen 37 pixels are separated by channel, and their HDR signals are averaged, obtaining the sky radiance for a sky point at the three channels. The uncertainty of these radiances is also calculated by the propagation of the HDR signal uncertainty of each pixel.

Figure 9 shows the extracted relative sky radiance at the three camera channels for one AERONET almucantar and one AERONET hybrid scan. The sky points have been marked in the left panels, clarifying the geometry of these scans. The obtained radiances are symmetric with respect to the Sun position, as expected under cloudless conditions, showing higher radiance and uncertainty values for the lower scattering angles (solar aureole). This symmetry with respect to the Sun is useful to discard cloud contaminated data (Román et al., 2017b). To compare with photometer measurements, we have extracted the

sky radiances at the same left and right points than the AERONET almucantar and hybrid scenarios, and both left and right radiance pairs have been averaged for each channel. The values showing differences above 20% between the left and right radiances have been classified as cloud contaminated and removed. This average and cloud-screening process, based on Holben et al. (1998, 2006), has been also applied to the AERONET sky radiances.

## 4 Results

### 4.1 Camera vs photometer

A comparison of the all-sky camera radiances with the measured photometer sky radiances has been done to evaluate the performance of the proposed methodology (Section 3). Camera effective wavelengths are not equal to the photometer but, in a first approximation, radiances at 440 nm, 500 and 675 nm has been assumed as at 467 nm (Blue), 536 nm (Green) and 605 nm (Red), respectively. The photometer and all-sky camera data used for comparison was recorded from July 2018 to March

2020. For each available AERONET almucantar and hybrid scenario, the closest HDR image within ±2.5 min window has been found, and the sky radiance at cloud-free points of the chosen scenario has been extracted from this image. For almucantar scans the point at azimuth equal to 180º has been discarded since the symmetry check for cloud screening cannot be applied. The camera radiance against photometer data is shown in Fig. 10 for each channel. The signal between camera and photometer radiances correlates with r values of 0.90, 0.88 and 0.80 for Blue, Green and Red channels, respectively. However, there are

several data pairs showing higher camera radiances than photometer ones. These deviated data correspond to scattering angles below 10º as indicated by the colour scale. Most of the data present a linear behaviour as shown by the density plots of Fig. 10, suggesting a linear relationship between radiances if the scattering angles below 10º are discarded. The correlation coefficients rise to 0.98, 0.98 and 0.97 for Blue, Green and Red channels, respectively, if scattering angles below 10º are not considered. The worse performance of the lowest scattering angles is not caused by problems in the linearity of HDR image,

especially for high signal values, because higher scattering angles show also high radiance values, but these ones fit well with the photometer measurements. Some observed light reflections in the camera dome and lens near the solar aureole image could be the main responsible of the obtained results for scattering angles below 10º. These angles will not be considered hereafter.

In order to compare camera and photometer radiances in a quantitative way, both radiances have been normalized. To this end, for each measurement group (almucantar or hybrid scenario) and channel, only the radiance points classified as cloud-free in both photometer and camera have been selected. Then, each radiance value of a given scenario and channel has been divided by the sum of all radiances at this channel. The sum of all normalized sky radiances at one channel must be equal to 1 for each measurement group. These normalized radiances give a relative information about sky radiance distribution and are useful to retrieve some aerosol properties (Román et al., 2017a). The correlation between camera and photometer normalized radiances is 0.99 for the three channels. Figure 11 presents the frequency histograms of the relative and absolute differences between the camera and photometer normalized radiances (excluding scattering angles below 10º). The difference distributions show a gaussian behaviour, with the maximum value around zero but slightly shifted to negative values except the absolute distribution of Red channel. The negative mean about -1.3% in the Blue and Green channels indicates a camera radiance underestimation of the photometer radiances at these channels. Red channel shows an overestimation by the camera about 3%, which is likely caused by the positive tail shown in the distribution. The mean of the absolute distributions is about 0. The standard deviation, associated to the uncertainty, ranges from 6.9% (Blue) to 13.2% (Red).

Figure 12 presents these differences as a function of scattering angle. These density-plots show higher density at specific angles because the AERONET hybrid scans are always done at the same scattering angles, while AERONET almucantar scans are measured at fix azimuth angles and therefore the scattering angle also depends on the SZA. In general, the camera radiances overestimate/underestimate the photometer ones for the lowest/biggest scattering angles for the Blue and Green channels. The differences for Red channel are around zero for the lowest angles, but the camera radiances strongly overestimate the photometer ones for scattering angles above 70º, especially in the almucantar scenario.

## 4.2 Camera vs simulations

Part of the obtained differences and this variation with scattering angle could be caused by the differences between camera and photometer wavelengths (e.g., in the Red channel this difference is about 70 nm). To solve that, sky radiances at the same camera effective wavelengths have been simulated with the radiative transfer model of the forward module of GRASP algorithm (Generalized retrieval of Atmosphere and Surface; Dubovik et al., 2014). The main inputs on this model have been obtained from AERONET retrievals: aerosol size distribution at 22 radii bins; real and imaginary refractive index at 440, 675, 870 nm linearly interpolated to camera wavelengths; and sphericity factor. Only AERONET retrievals satisfying an inversion error below 10% in the sky radiances have been used. Valladolid climatological values of the bidirectional reflectance distribution function (BRDF) parameters, obtained from MODIS MCD43C1 (Schaff et al., 2011) satellite product (see Román et al., 2018), have been interpolated to camera wavelengths and used as input in GRASP. With this information the radiative transfer model is capable to simulate the sky radiance at any desired sky direction; in our case these points have been selected to match the AERONET almucantar and hybrid scans, separately.

First, the performance of the radiative transfer simulations has been evaluated against photometer data, by simulating sky radiances at 440m 500 and 675 nm. The used data are from July 2018 to March 2020 and each simulated scan has been directly

compared with the temporal closest photometer cloud-free radiance scan. It means that we are comparing the radiance obtained by certain aerosol properties with the sky radiance used to retrieve those aerosol properties. In this case, the simulated and measured radiance agree with r values of 1.00 for the three wavelengths, and without any dependence on scattering angle, even for scattering angles below 10º (not shown). The differences between simulated and photometer normalized radiances (rejecting scattering angles below 10º) shows a mean value about 0.2%, 0.4% and 0.7%, and a standard deviation about 2.3%, 2.8% and 3.2%, for 440, 500 and 675 nm, respectively. These results are within the nominal calibration uncertainty of the AERONET radiance measurements.

The radiance at almucantar (which is measured only for SZA>40º) and hybrid points have been separately simulated for the HDR sky images that are closest in time (within 10 minutes) to the available aerosol AERONET retrievals (with sky error below 10%) from July 2018 to March 2020. A direct comparison between camera and simulated radiances (not shown) presents worse agreement for scattering angles below 10º, but also reveals higher r values: 0.94, 0.92 and 0.91 for 467, 536 and 605 nm, respectively; these r values rise to 0.99, 0.99 and 0.98 when scattering angles below 10º are not considered, and to 0.99 for all wavelengths when the radiances are also normalized.

Figure 13 shows the distribution of the normalized radiance differences between camera and simulations. These distributions also show a Gaussian behaviour, as it was observed in the camera-photometer comparison. However, the mean and standard deviation are significantly lower. The mean values are about zero indicating no over- or underestimation, and the standard deviation values reveal an uncertainty on camera radiances about 5.2%, 7.3% and 9.9% for 467, 536 and 605 nm, respectively. This observed improvement in the correlation and in the mean and standard deviation of the obtained differences, points out the influence of the wavelength differences on the camera-photometer comparison.

The left panels of Fig. 14 represent the dependence on scattering angle of the camera-simulated differences in normalized radiance. The behaviour is similar than the one obtained using photometer measurements instead of simulations, except for 605 nm where the camera also overestimates the simulations for the lowest angles. High difference values appear from 80º to 120º scattering angles, especially at 605 nm and almucantar scans, which is also observed in Fig. 12. These differences appear for points with zenith angles from 48º to 65º, which corresponds with the position of the observed ring image reflected on the dome (see Figs. 2 and 9); this is an image of a piece of the prototype camera that lost part of its black colour. Right panels of Fig. 14 show the differences for normalized radiances computed without the points in the 48º-65º zenith angle range. The high differences previously observed around 80º-120º scattering angles disappear if the mentioned observation angles are not considered. Under these conditions (not zenith angles from 48º to 65º), the normalized camera radiances show a slight overestimation on the simulated values for scattering values below 15º. The dependence on scattering angle is stronger in 605 nm, while for 467 nm this dependence is not clear except for the lowest angles.

The mean of the differences between camera-simulated radiance is between -0.6% and -0.1% for the three channels when the points affected by the reflection in the dome are discarded; for these conditions the standard deviation is reduced to 4.4%, 5.7% and 6.9% for 467, 536 and 605 nm, respectively. For the same conditions (not scattering angles below 10º neither zenith points between 48º and 65º) but  applying a stronger cloud-free threshold of 5% instead of 20%, the number of available data

is reduced to 86%, 80% and 74%, but the standard deviation goes down to 3.5%, 4.5% and 5.6% for 467, 536 and 605 nm, respectively, while the mean difference values are still close to zero (between -0.4% and -0.1%) for the three channels. Under these conditions, the 74% (467 nm), 67% (536 nm) and 64% (605 nm) of the obtained camera-simulated radiance differences are within the combined uncertainty associated to the camera and simulation values; these percentage of data rises to 93%, 89% and 88% for the expanded uncertainty (the double of the combined uncertainty). Both obtained values are about 69% and

95%, which are the expected values for a gaussian distribution with standard deviation equal to the uncertainty. This result indicates that the uncertainty associated to the camera radiances could be representative of the real camera uncertainty if the problematic camera angles are not used.

Finally, if the camera radiances with a propagated uncertainty above 5% are also rejected (less than 0.5% of total data), the differences on camera-simulated radiances present mean values ranging from -0.3% to -0.1% and standard deviation values of

3.3%, 4.3% and 5.3% for 467, 536 and 605 nm, respectively. The rejection of camera radiances with a propagated uncertainty (inherent to both HDR image calculation and radiance extraction method) above 5% also reduces the number of data outliers observed for scattering angles below 10º. In this sense, the differences on the camera-simulated radiances have been calculated applying all the mentioned quality criteria (rejection of points viewing the camera reflected image; cloud-free within 5% in symmetric points; required camera radiance uncertainty below 5%) but rejecting only scattering angles below 8º, 7º and 6º. As

result, for 467 nm, 536 and 705 nm the standard deviation of the differences are: 4.0%, 5.0% and 5.8% (with mean values between -0.7% and -0.5%) discarding angles below 8º; 4.5%, 5.4% and 6.1% (with mean values between -0.9% and -0.7%) discarding angles below 7º; and 5.5%, 5.8% and 6.5% (with mean values between -1.1% and -1.0%) discarding angles below 6º. In this case, camera radiances overestimate the simulated ones at the lowest angles and underestimate the simulations for the rest of angles; this fact leads to the observed increase (in absolute value) in the mean camera-simulations differences.

Therefore, the use of normalized sky radiances of this work is more appropriated without the scattering angles below 10º. However, sky radiance at low scattering angles could be useful for some purposes such as aerosol retrieval. The uncertainty estimates provided here should be considered in the retrieval.

## 5 Conclusions

The present work proposes a new methodology to obtain the relative (normalized) sky radiances and their uncertainty from

all-sky camera images. To this end, an all-sky camera (SONA202-NF), equipped with three spectral channels (narrower than usual) and with effective wavelengths of 467, 536 and 605 nm, has been used in this paper. The proposed method only requires sky images and a set of camera images under dark conditions, both at various exposure times. Dark frames are useful to characterize the camera readout noise and black level, but any white balance correction must be avoided to this end. Hot pixels can be detected through the correlation between the pixel dark signal and temperature. The linear response of pixel signal can

be characterized by taking images with different exposure times; the previous knowledge of these exposure times is not necessary since the ratio between them can be calculated as the slope of a linear fit between the pixel signals at two different

exposure times. These slopes give a relationship between exposure times that provide an effective linear response for the sensor. The characterization of these parameters has allowed the calculation of a linear HDR image and then a relative sky radiance map and its uncertainty at the three camera spectral channels. The relative sky radiance at any sky direction can be

extracted from these maps.

The relative sky radiance obtained by the proposed method has been compared with the radiances measured by a CE318-T photometer at the closest wavelengths 440, 500 and 675 nm. Both radiances agree for the three wavelengths except for scattering angles below 10º, which could be mainly caused by solar light reflections in the fisheye lens and camera dome near the Sun position. The distribution of the differences between normalized sky radiances have shown standard deviations from

7% (467 nm) to 13% (605 nm), being part of these differences caused by the differences on both instrument wavelengths.

To solve the wavelength shift between instruments, the camera radiances have been compared against radiance simulations at the same wavelengths, using the AERONET aerosol properties as input in a radiative transfer model. This comparison reveals an uncertainty on normalized all-sky camera radiances, with scattering angle above or equal to 10º, about 5%, 7% and 10% for 467, 536 and 605 nm, respectively. However, this uncertainty is reduced to 3.3% (467 nm), 4.3% (536 nm) and 5.3% (605

nm) if the following quality criteria are applied: rejection of radiances under scattering angles below 10º; radiance assumed as cloud-free only when left-right symmetric data pairs show differences below 5%; exclusion of radiances with a propagated uncertainty above 5%; rejection of radiance values with zenith angles between 48º and 65º, which encompasses an area contaminated by a reflected image of part of the camera. The normalized camera sky radiance slightly overestimates the simulations at the lowest scattering angles.

With the obtained results, we make two recommendations to all-sky camera manufactures: 1) the application of white balance should be done after raw image capturing instead of before, because it avoids unnecessary pixel saturations and reduces the shot noise; and 2) the reduction of reflected images in the fisheye lens and camera dome which can contaminate the sky radiance maps. The spectral filter width reduction on the SONA camera filters allows the use of all-sky cameras for novel approaches; narrower filters would be more helpful in the future, since in the current filter setup, some colour channels are still

sensitive to wavelengths associated to the other channels.

The obtained camera relative sky radiances could be calibrated in absolute physical units, but this was out of the scope of this paper. The determination of normalized sky radiances is useful for the retrieval of aerosol properties. Therefore, we will try to use in the future this kind of measurements in combination with other instruments to retrieve aerosol properties like the particle size distribution. Finally, we encourage other researchers interested in sky radiance data, to apply the developed method on

their all-sky cameras to obtain relative sky radiances maps.

**Acknowledgements**

The authors thank the Spanish Ministry of Science, Innovation and Universities for the support through the ePOLAAR project (RTI2018-097864-B-I00). The authors acknowledge the use of GRASP inversion algorithm software (https://www.grasp-

open.com) in this work and also thank the team of GRASP SAS by the support and training on GRASP code. LibRadtran
developers are also acknowledged for the use of their code. The authors gratefully thank AERONET for the used aerosol
products. Finally, the authors thank the GOA-UVa staff members who helped with the maintenance task of the used
instruments and the support of station infrastructure.

## Author contribution

JCAS and RR designed and developed the main concepts and ideas behind this work and wrote the paper with input from all
authors. They also run the different irradiance/radiance simulations. CL developed and provided the used all-sky camera. RG
was responsible for the camera being operational at the Valladolid station. VEC and CT were responsible of the used Valladolid
AERONET station. DM, AC and AMdF contributed in the interpretation of results.

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

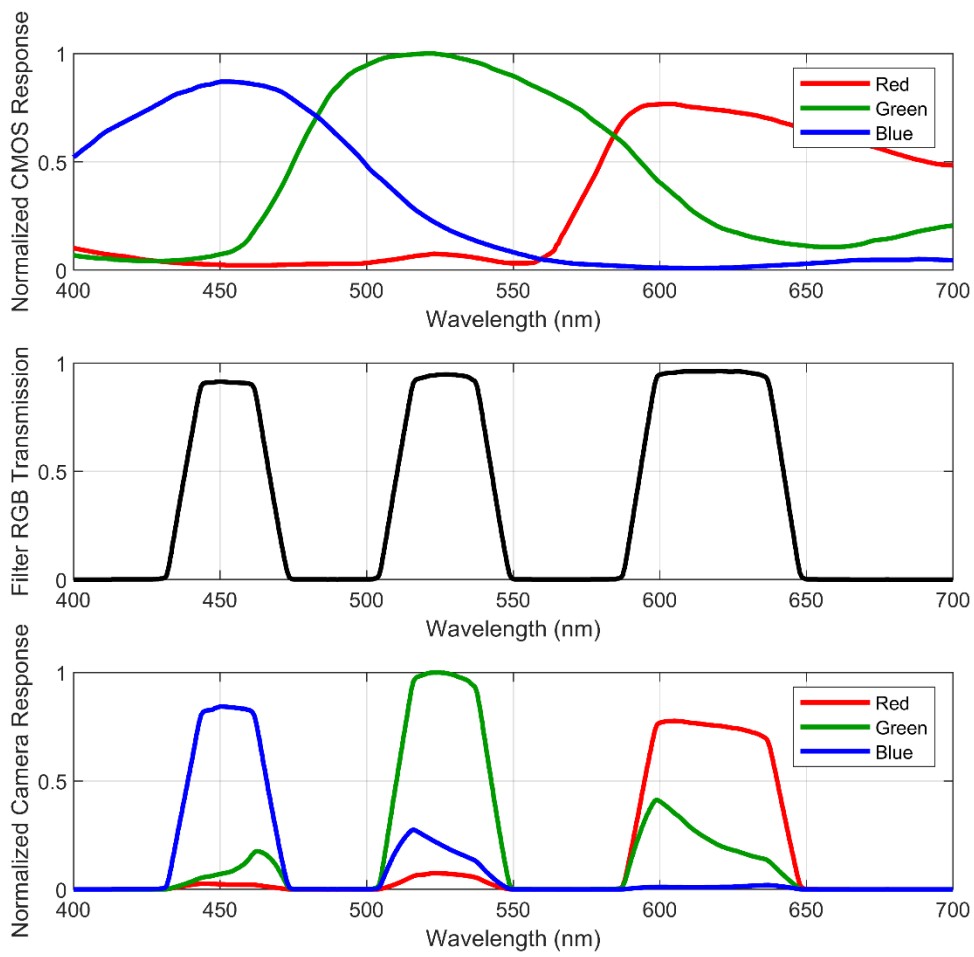

**Figure 1: Spectral response of: a) CMOS sensor Bayer filters; b) RGB triband filter; and c) the all-sky camera (both filters together).**


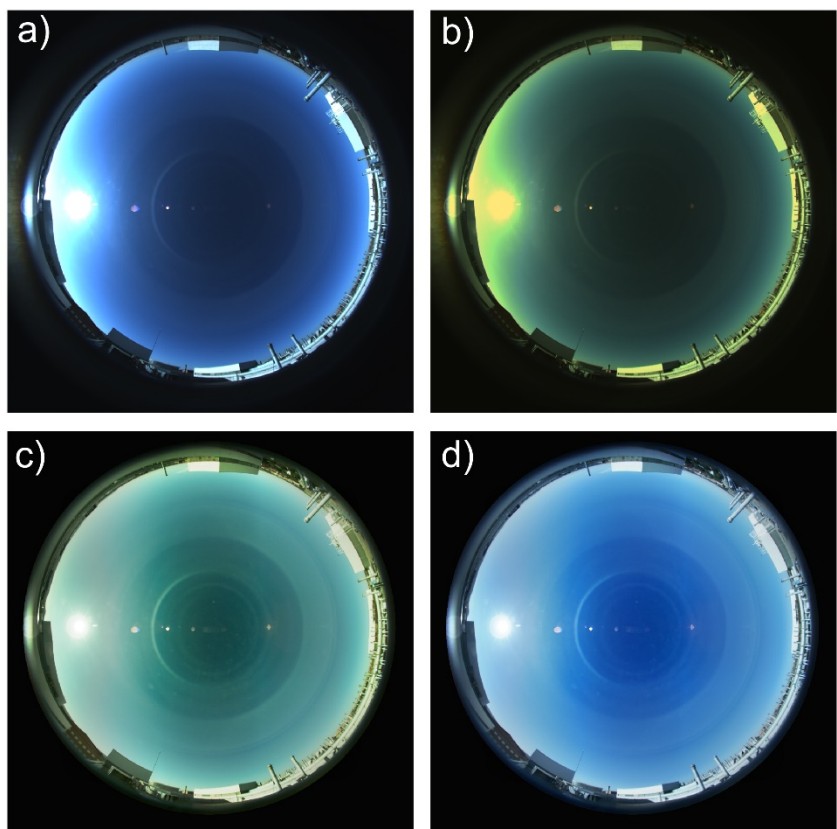

**Figure 2: Colour sky image of Valladolid at 17 August 2019 07:25 UTC of the a) direct capture with exposure time equal to $t_4$; b) direct capture with exposure time equal to $t_4$ but removing the previous white balance; c) tone map of HDR image; and d) tone map of HDR image but with white balance applied.**




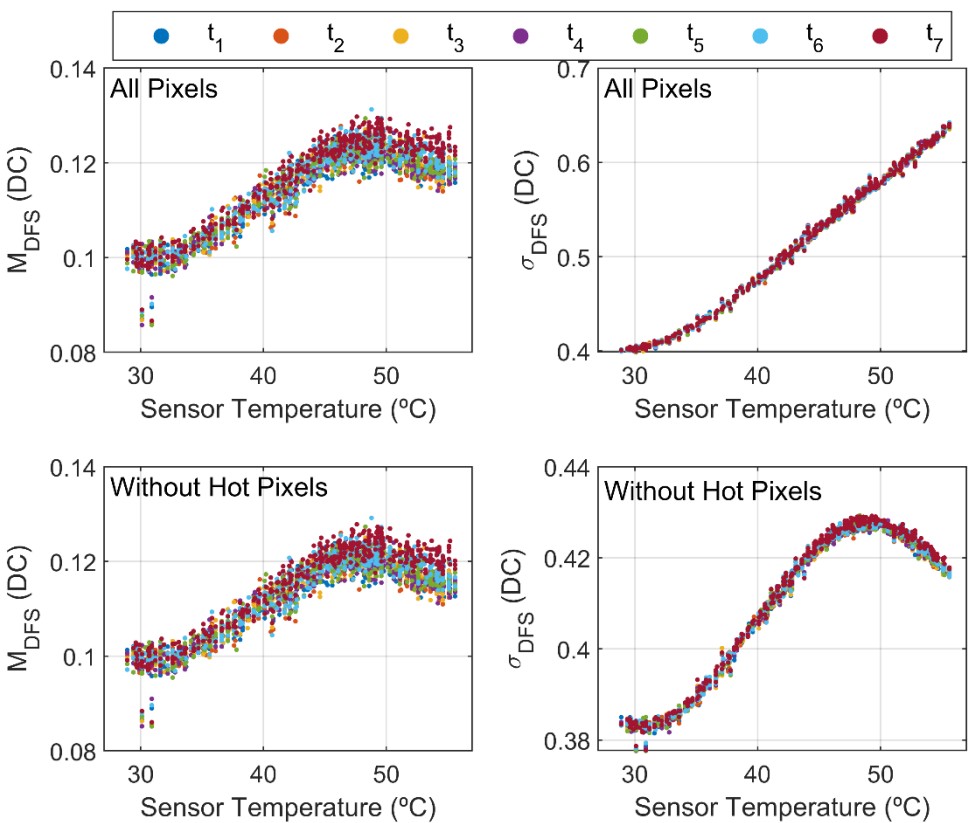

**Figure 3: Mean (left panels) and standard deviation (right panels) of the dark frame signal as a function of sensor temperature for different exposure times. Values on the upper panels are calculated considering all pixels and bottom panels without hot pixels.**

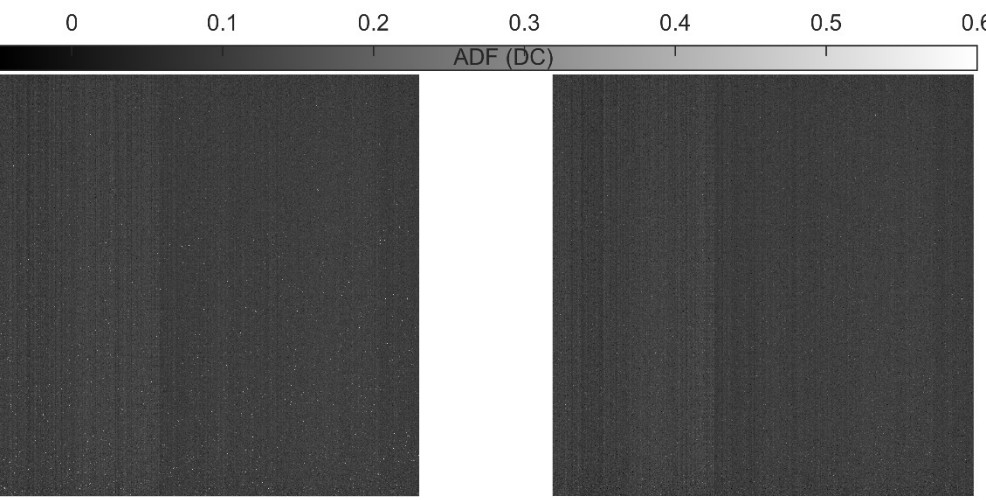

**Figure 4: Averaged dark frame with (left panel) and without (right panel) hot pixels.**



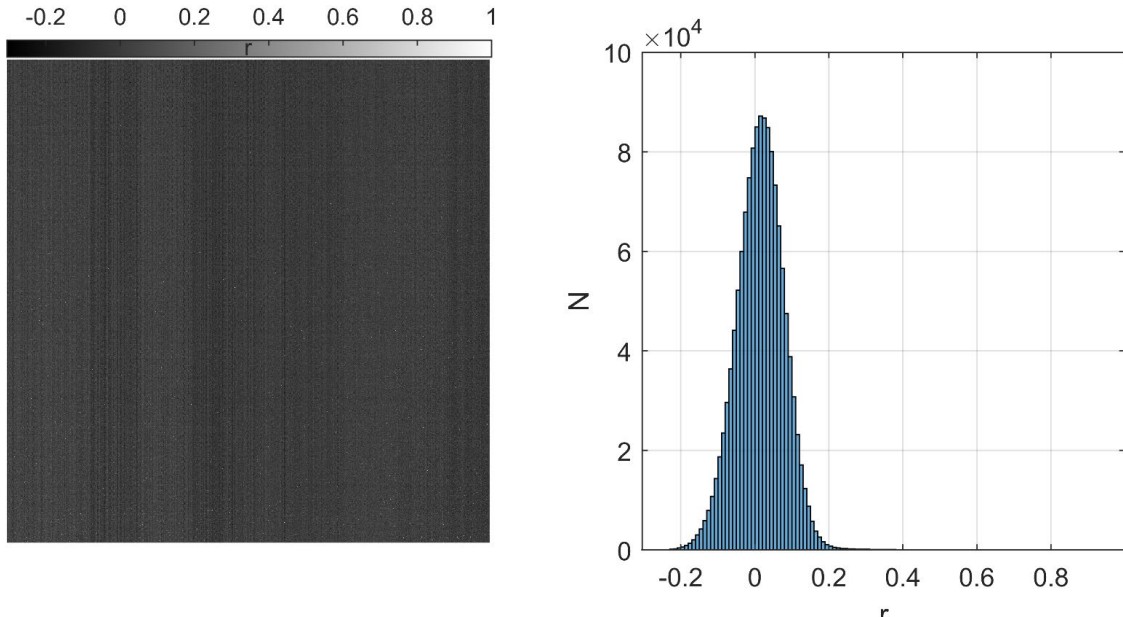

**Figure 5: Correlation coefficient of pixel signal at $t_1$ with temperature for each pixel (left panel) and its frequency distribution (right panel).**


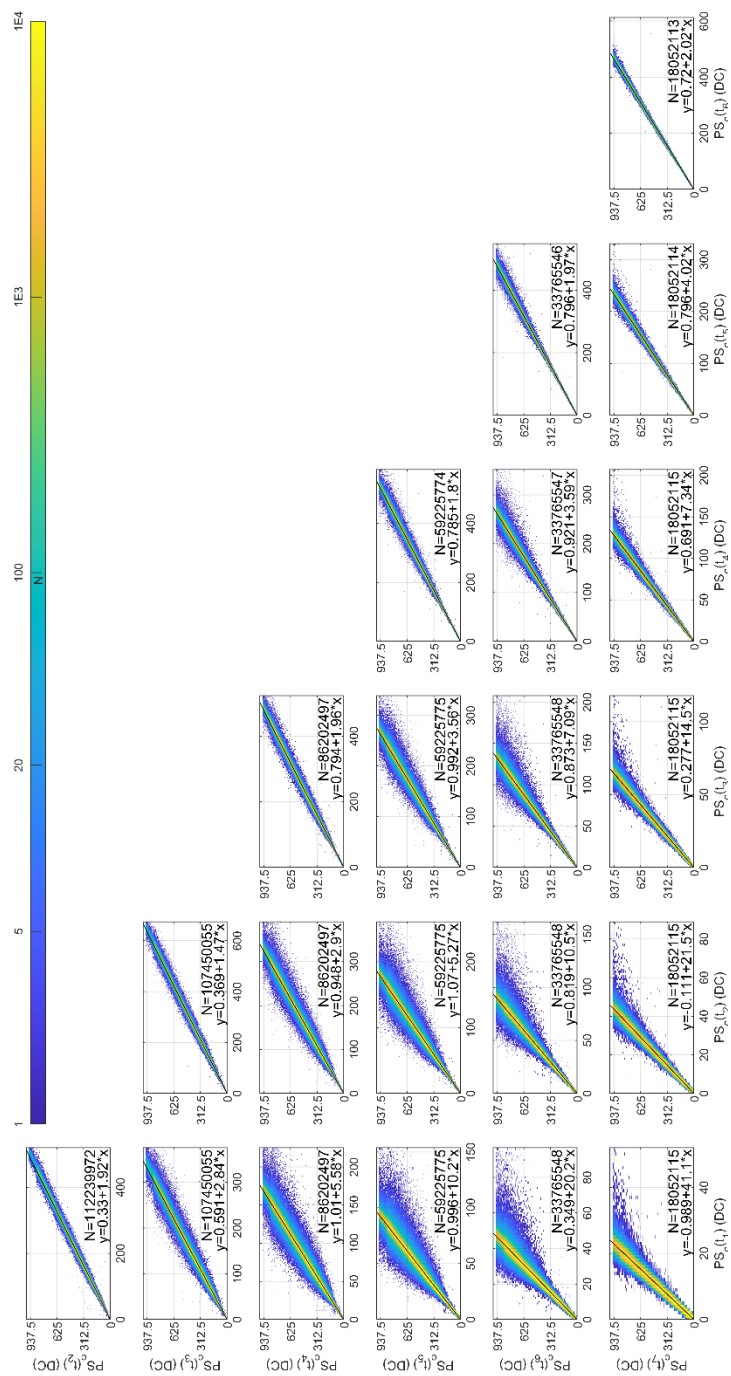

**Figure 6: Corrected pixel signal at different exposure times as a function of the corrected pixel signal at other exposure times for all available daytime images on 18 August 2019 at Valladolid. Saturated pixels are not included. The panels show the weighted least square linear fit and the number of data (N).**

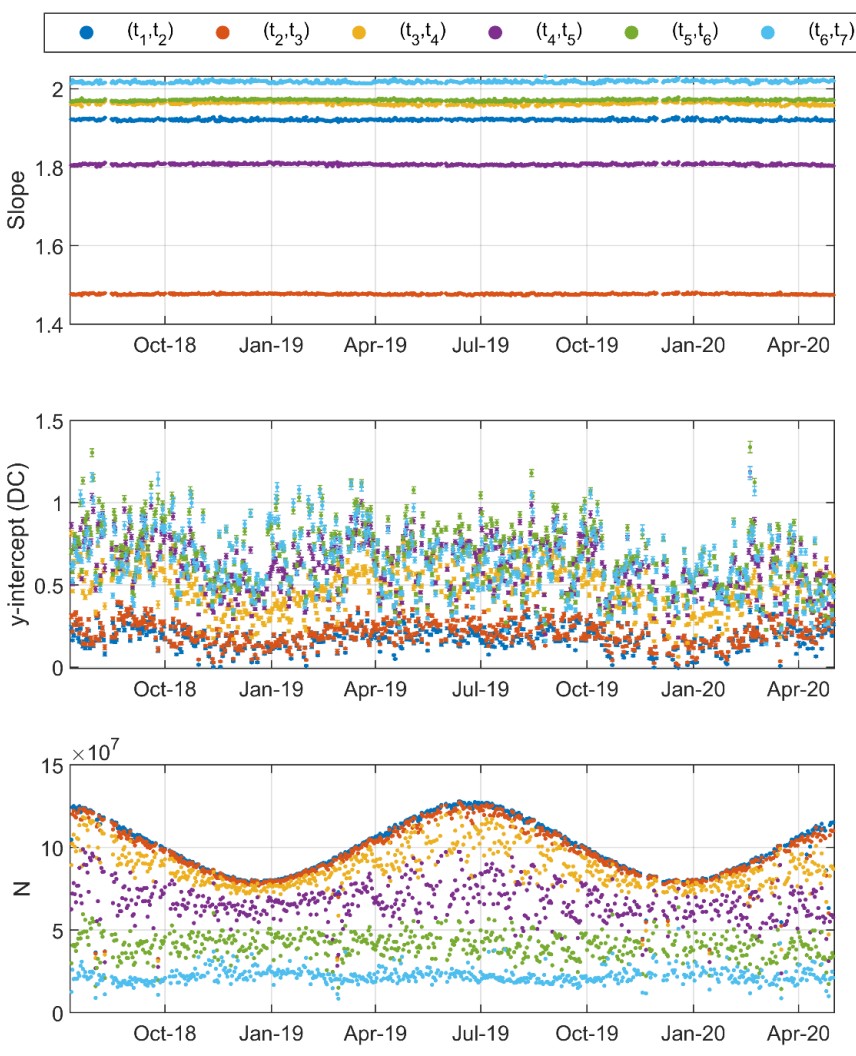

**Figure 7: Time evolution of the a) slope, b) y-intercept and c) number of used data (N) of the daily weighted least squares linear fits for different exposure time pairs.**


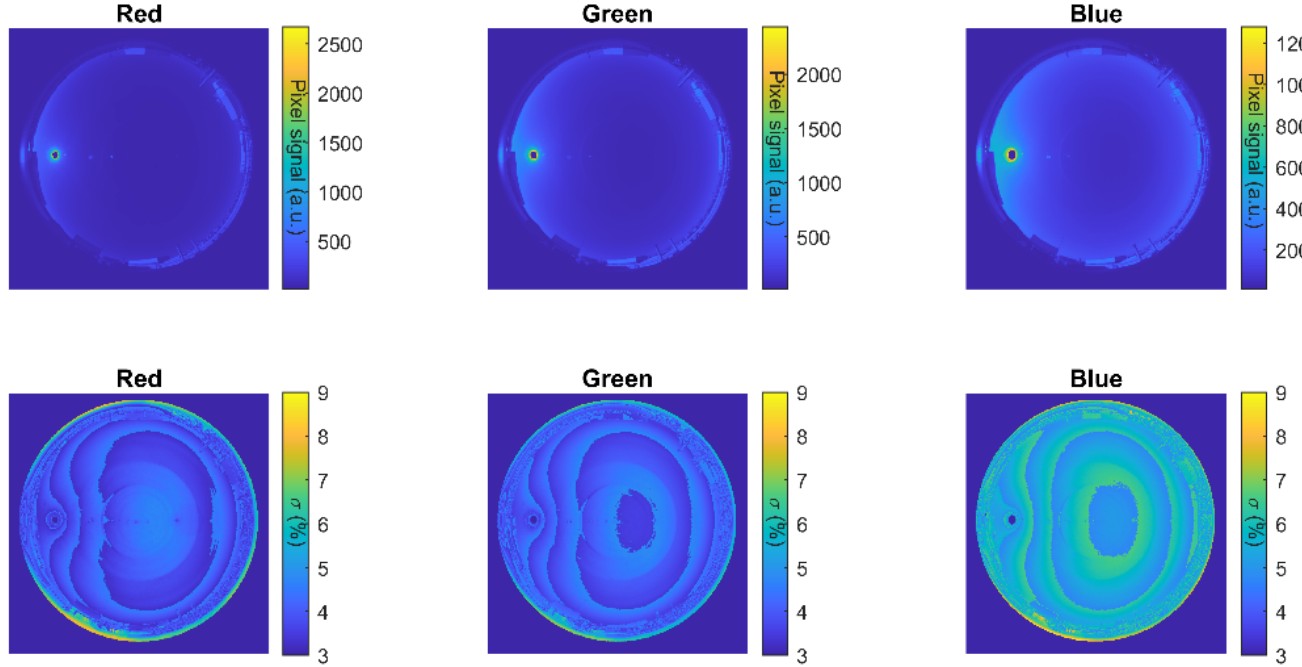

**Figure 8: Linear HDR pixel signal (upper panels) and its uncertainty (bottom panels) for the Red (left), Green (middle) and Blue (right) channels at Valladolid on 17 August 2019 07:25 UTC.**



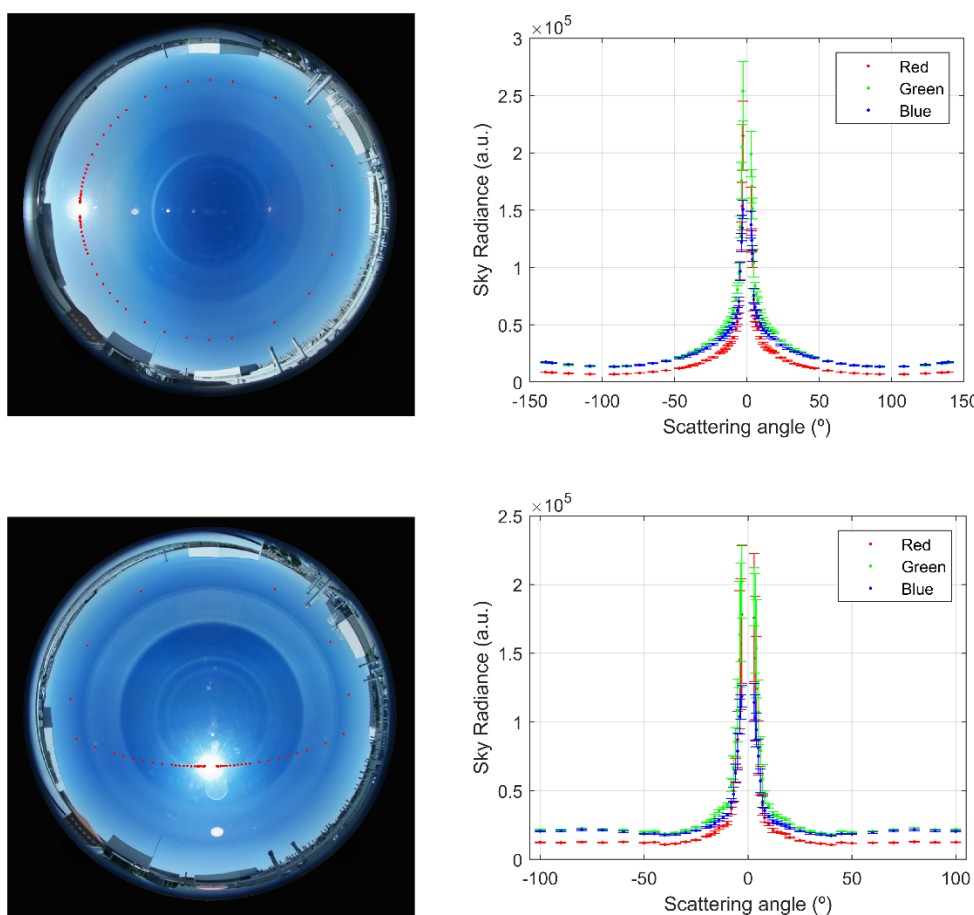

**Figure 9: Camera sky radiance at the three channels (right panels) for almucantar (upper panel) and hybrid (bottom panel) scans on 17 August on 07:25 UTC and 12:25 UTC, respectively. Left panels show in red the almucantar and hybrid sky points on a tone map of the demosaiced HDR image.**




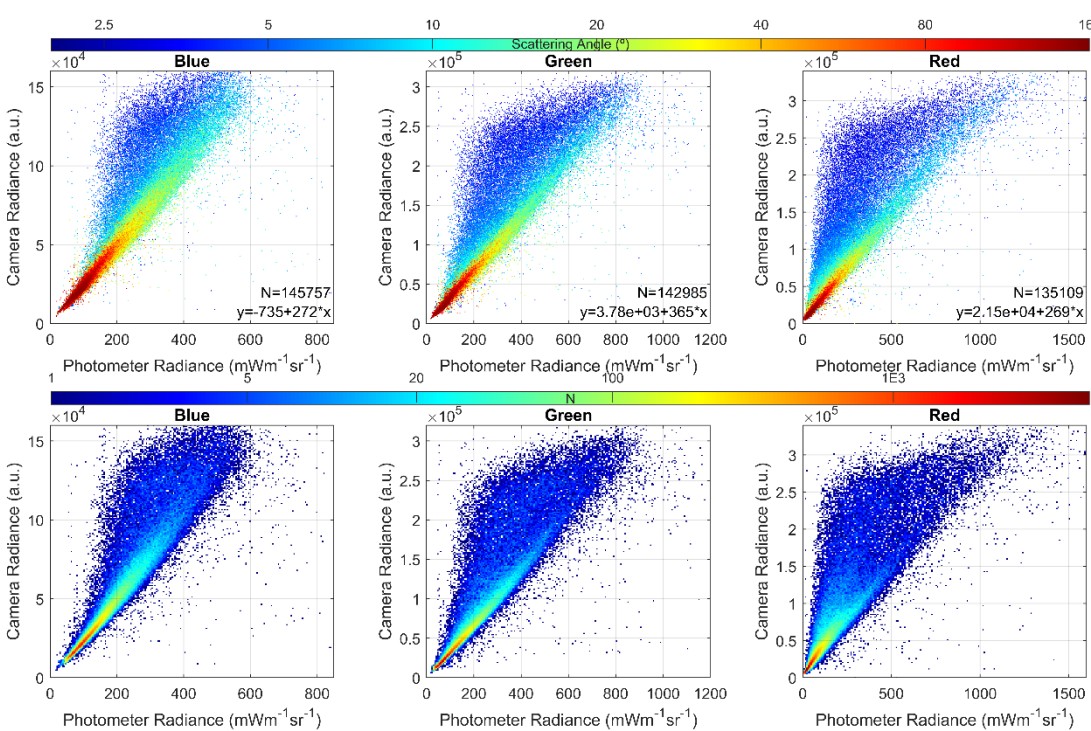

**Figure 10: Camera sky radiance as a function of photometer radiances for Blue (left), Green (middle) and Red (right) channels, respectively. Colour scale represents the scattering angle in upper panels, while it means the number of available data N in an interval (density plot) in bottom panels.**



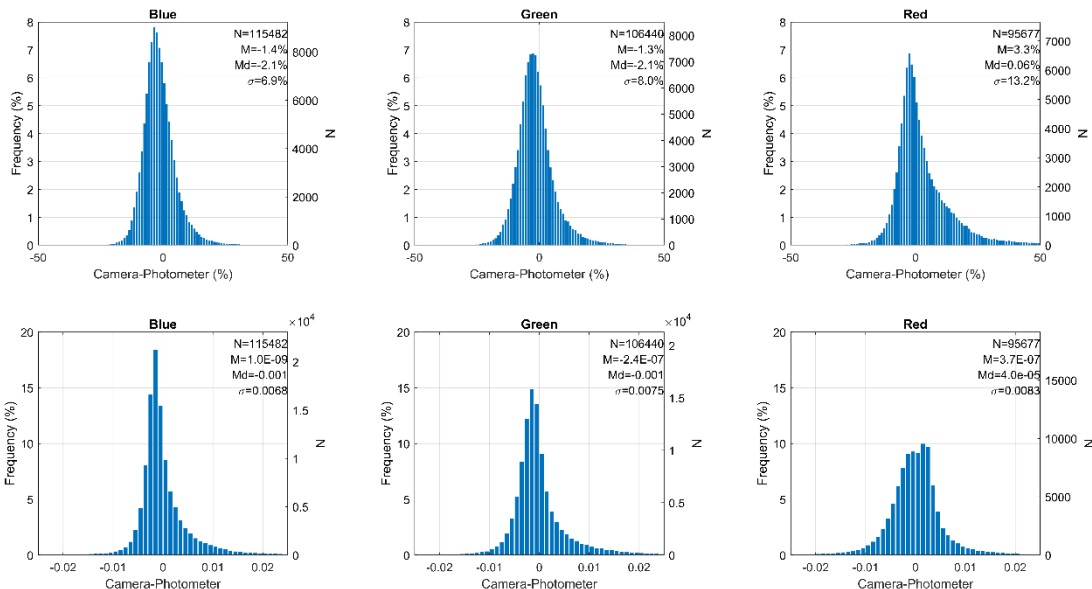

**Figure 11: Frequency histograms of the relative (upper panels) and absolute (bottom panels) differences between the camera and photometer normalized sky radiances for Blue (left), Green (middle) and Red (right) channels, respectively. Radiances with scattering angles below 10º are not considered. The number of data (N), and the mean (M), median (Md), standard deviation (σ) of the differences are also included.**


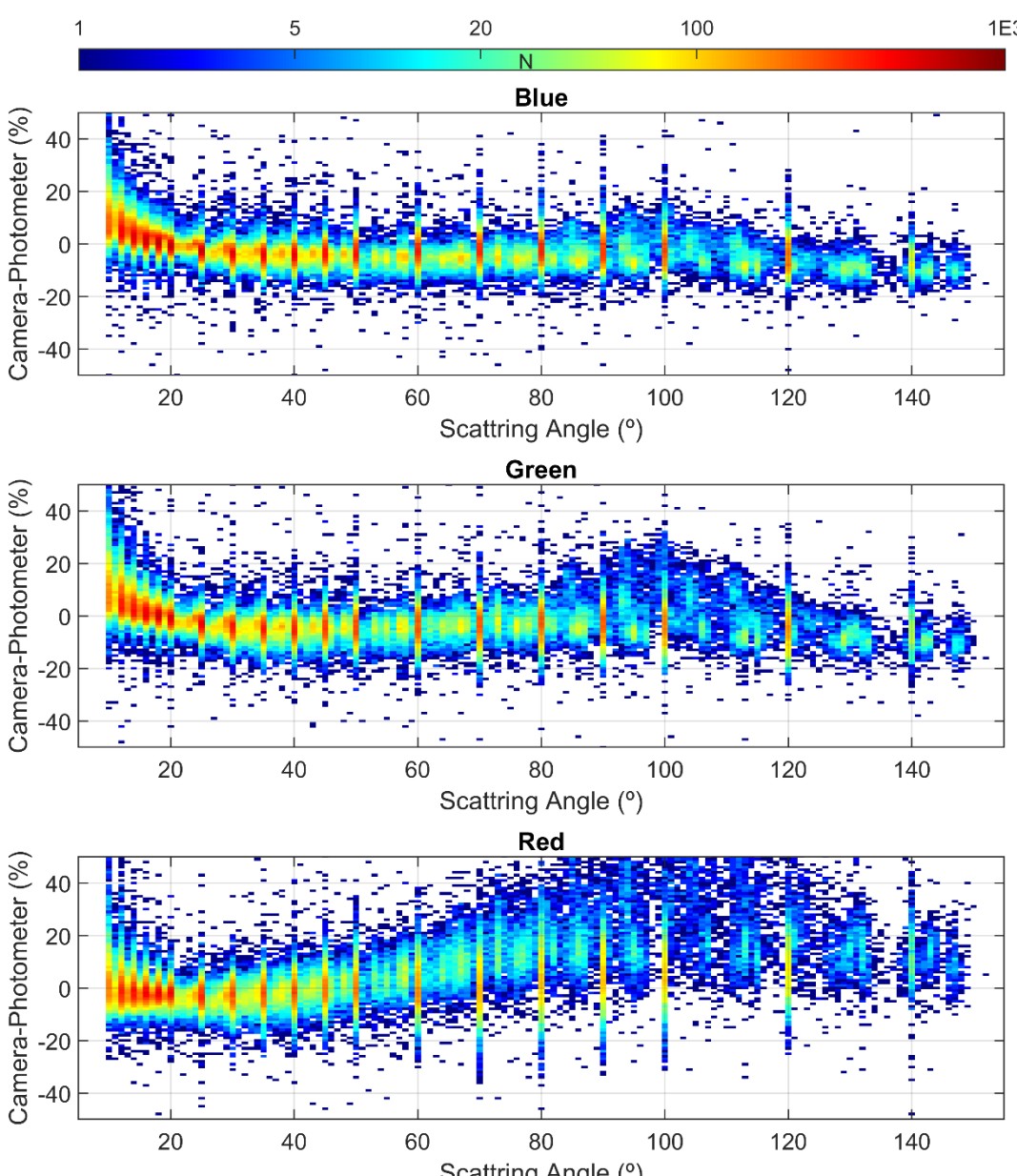

**Figure 12: Relative differences between the camera and photometer normalized radiances as a function of the scattering angle for Blue (upper panel), Green (middle panel) and Red (bottom panel) channels, respectively. Radiances with scattering angle below 10° are not considered.**

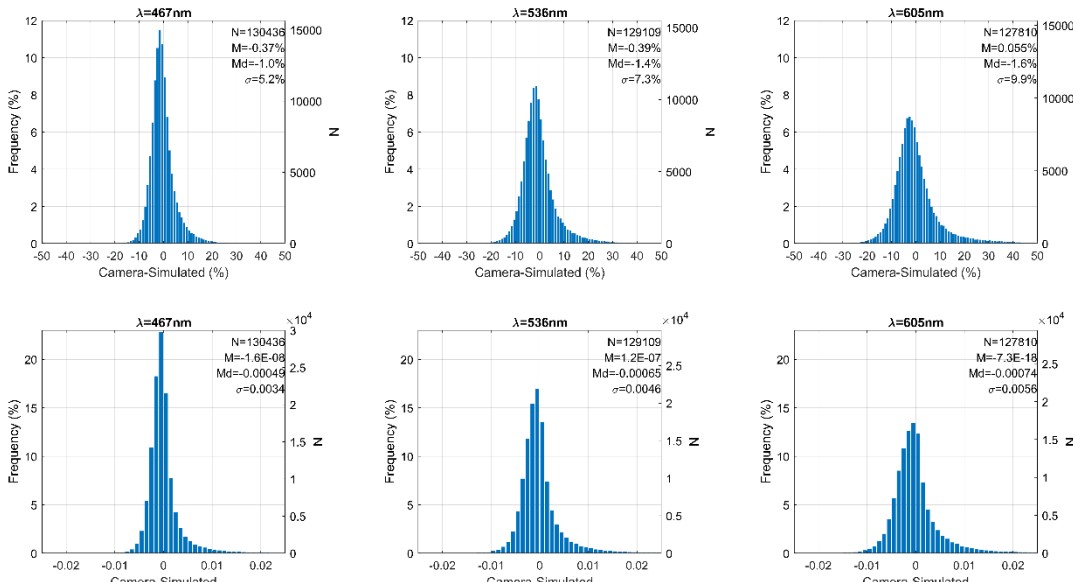

**Figure 13: Frequency histograms of the relative (upper panels) and absolute (bottom panels) differences between the camera and simulated normalized sky radiances for 467 nm (left), 536 nm (middle) and 605 nm (right), respectively. Radiances with scattering angles below 10º are not considered. The number of data (N), and the mean (M), median (Md), standard deviation (σ) of the differences are also included.**



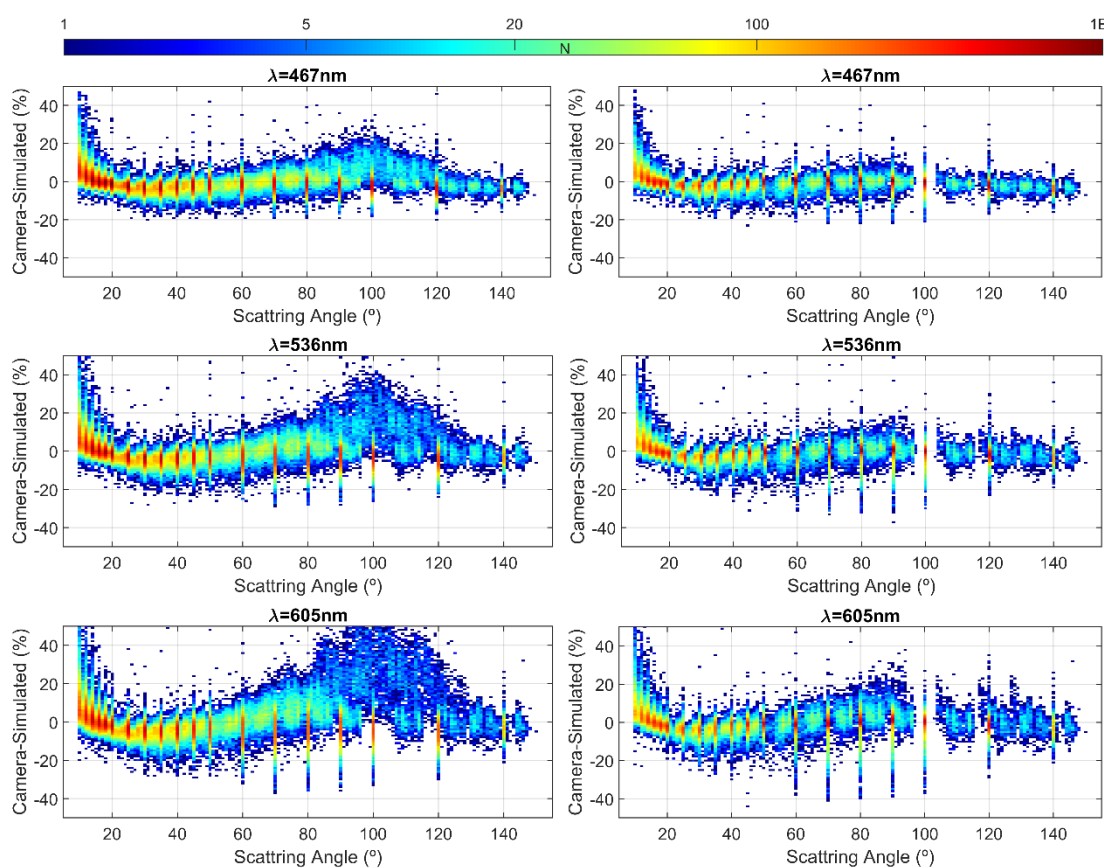

**Figure 14: Relative differences between the camera and simulated normalized radiances as a function of the scattering angle for 467 nm (upper panels), 536 nm (middle panels) and 605 nm (bottom panels), respectively. Left panels show the differences calculated without radiances under scattering angle below 10º, while right panels show the same differences but also obtained without radiances under zenith angles between 48º and 65º.**


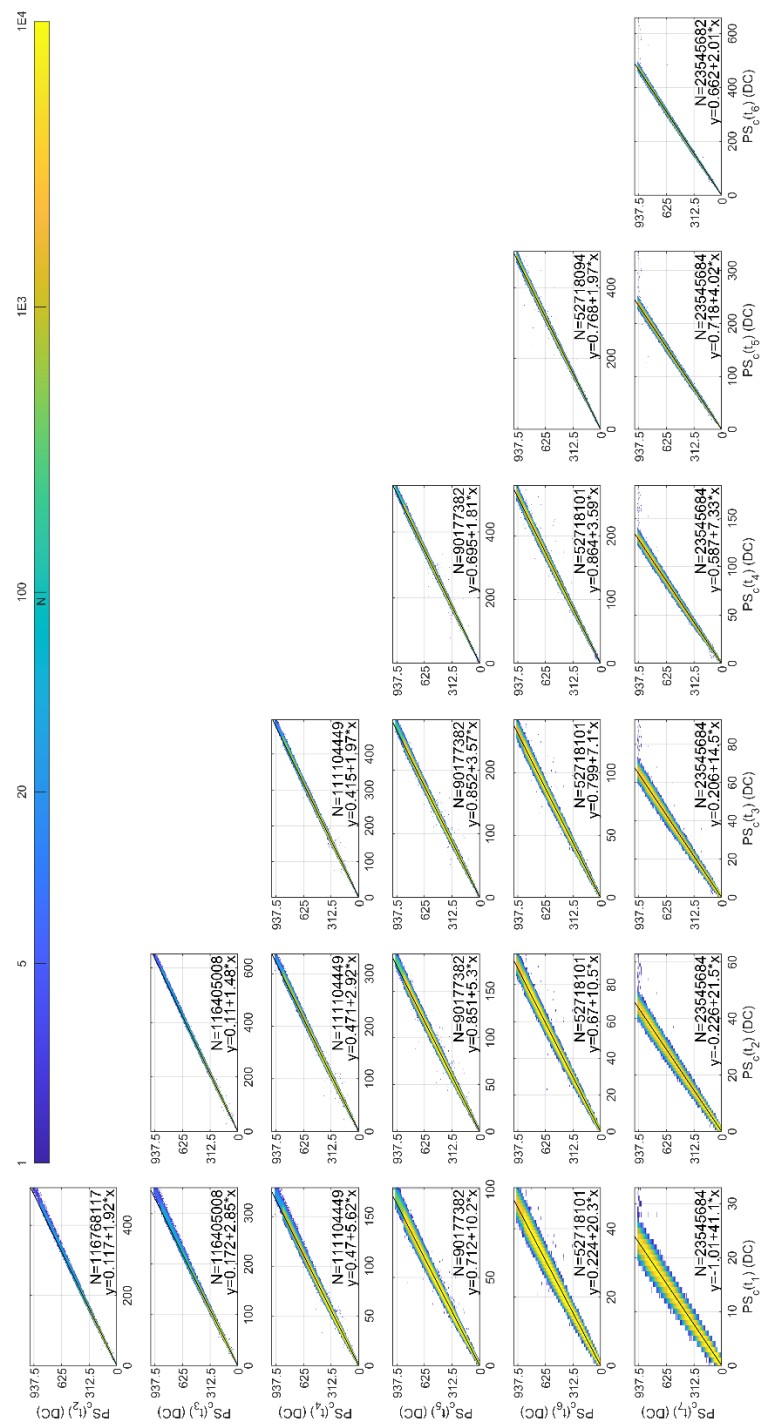

**Figure A1: Corrected pixel signal at different exposure times as a function of the corrected pixel signal at other exposure times for all available daytime images on 17 August 2019 at Valladolid. Saturated pixels are not included. The panels show the weighted least square linear fit and the number of data (N).**