# Peer review of "Relative sky radiance from multi-exposure all-sky camera images"

_Atmospheric Measurement Techniques, 2020_

## Referee Comment (RC1) · Anonymous Referee #1 · 2 Dec 2020

This manuscript presents a generally well-written study on the relative radiance calibration of a sky imager using HDR images. The study presents a comprehensive method in order to take into account all possible sources of uncertainty in obtaining the HDR image and, finally, the relative sky radiance. I recommend this study for publication in AMT after some minor revisions.

On section 2, it is not specified the origin of the spectral response of the camera filters shown in Fig. 1. Is it from the data sheet from the CCD manufacturer? Are they calculated somehow? Please specify. The exact setup of the RGB triband filter is not clear. How are these filters coupled to the camera body or CMOS sensor? In addition, the same as before, where does the spectral response of the filters come from?

On section 3.2, what I'm understanding here is that the camera provides the images

with a white balance gains of 1.1 and 2.1 for the G and B channels respectively. The white balance is reversed by dividing the pixel values by these gains (as in Eq. 1) and the everything is calculated with the image without the white balance, correct? If that is the case, the effect of having saturated pixels due to the white balance still exist. Is it possible to manually set the white balance off (or white balance gains to 1) in the camera?

On section 3.3 (paragraph starting on line 173), the M_DFS and SIGMA_DFS are average and standard deviation of the sum of the three channels? Or is it one channel (which one?)

On the same section, finally, a temperature correction of the dark signal is not applied, right?

On section 3.4, why the cloudy day (18th August) is used instead of a clear day (the 17th) to show the signal value at one exposure time vs other exposure times? Why adding an Fig. A1 when that could the Fig. 6? In addition, on Fig. A1, the description says 18th august and should be 17th.

On page 8, line 233, after readout noise it should say "(N_r)" just for clarification.

On section 4.1 it is introduced that the time window for comparison is 10 minutes. Having almost 2 years of data, it might be possible to narrow this window. This could probably have an impact on the deviation of radiance values at small scattering angles (<10°) besides other possible effects. As shown on Fig. 9, the slope of the radiance for small scattering angles is very steep, and a difference in sun position between image and photometer might have an impact. At least it could be quantified.

As a final comment, it if very surprising that the effect of the reflection of the lens on the dome is not higher. It is clearly visible on the images and, even though the almucantar and hybrid configurations explore very specific angles, I would expect a higher impact, especially for the hybrid configuration where the angles measure pass transversally the

reflection (as shown on Fig. 9).

---

## Referee Comment (RC2) · Anonymous Referee #2 · 7 Dec 2020

General comments:

The manuscript describes the characterization of an all-sky camera to extract relative sky radiances which are then compared to measured (AERONET) and modelled radiances.

Although no real new technique is revealed in the manuscript, the topic is relevant for atmospheric science and fits the scope of the journal. I would suggest a number of improvements, which add up to a major revision, to raise the overall quality and enhance the readability.

Specific comments:

L11pp. Please consider to improve the description of the relative / normalized radiance

[Figure]

- see also the respective comment below.

L14. "in line" is a little loose phrasing for a scientific abstract.

L30pp. Please rewrite the first two paragraphs in a more precise way and better English language. For example, . . . responsible for the colour of the blue sky under clear conditions, and also by aerosols and surface albedo (rather than clouds, in clear conditions, I guess).

L53. "to derive sky radiance measurements..." It would be good for the introduction to state what is new/improved in the current work in relation to that past work.

L92. Are the sky radiance available at the AERONET website? If yes, where exactly?

L105. How many pixels are the threshold here?

L137pp. If white balance is about the relative scaling of the channels, how does it affect the individual relative radiances, which is the subject here? If the answer is "it does not", please justify the inclusion of this section/topic.

L152pp. Please clarify this section. In general, the measured signal is a (non-) linear function of the number of absorbed photons plus a dark signal offset plus a noise, right? So dark signal (offset) and noise should be distinguished.

L175pp. Is there any explanation for the effect of the decrease again for temperatures above 50°?

L179pp. The benefits of Figure 4 and 5 (left) are questionable. I am afraid, I can only see black squares.

L202pp. Is there any less confusing way of presenting these numbers for the various cases?

L216pp. I find this explanation and the corresponding figure 6 quite confusing to read. Please try to improve here. Also avoid the "jet" or rainbow type colormap, as it is known

to e.g. not be perceptually uniform. Maybe use the colormap of Fig. 8 throughout.

L301pp. Maybe clarify the nomenclature of "relative" radiance (relative to what?). Would it be more accurate to call it "corrected signal"? Later, the radiances are normalized relative to the sum of radiances.

L329. The concept of "dispersion" is usually applied differently.

L334pp. Is the word "linearity" used in two different ways in the same sentence?

L369pp. This whole paragraph is not really relevant for the paper (validation of the model?), nor is the conclusion surprising (r=1 for the comparison).

L383. "when the radiances are also normalized." It seemed that the radiances should always be normalized for all comparisons.

L388. Would the wavelength difference also affect the standard deviation?

L423pp. Please rewrite this sentence in a more comprehensible way.

L575. Fig. 2. The letter annotation in the panels is missing.

L645. Fig. 9. There are very noticeable concentric rings in the images (same in Fig. 2), the jumps are even visible in the lower radiance plot (around +/- 40° scattering angle). This looks like a major issue for the topic at hand and should be discussed accordingly. While all-sky images typically show some kind of internal reflections and artifacts, these look quite a bit worse than in many examples I have seen from other cameras.

Technical corrections:

I acknowledge that the authors are not native English speakers, but please make an effort to improve the language in general, in terms of grammar (even spelling) and clarity. Some examples below:

L12 maps.

[Figure]

L14. Except for.

L42. "Earth's" or "the earth", "the atmosphere".

L86. "measure" sky radiances.

L136. "a wider".

L138. "affects the".

L140. "multiplication of the recorded"

L306. "radius of 3"

L286. "as can be seen in Fig. 2."

Fig.12 x-labels: scattering angle. (same in Fig. 14)

———————————————————

---

## Author Comment (AC1) · 14 Jan 2021

**Response to the Referee #1 comments for the manuscript "Relative sky radiance from multi-exposure all-sky camera images" By Juan Carlos Antuña Sánchez et al. in AMTD**

First, we are grateful for the effort of Referee #1 and her/his review in detail. Reviewer comments are in black font (RC), and author comments (AC) in red font.

**Author's answer to Anonymous Referee #1**

RC: This manuscript presents a generally well-written study on the relative radiance calibration of a sky imager using HDR images. The study presents a comprehensive method in order to take into account all possible sources of uncertainty in obtaining the HDR image and, finally, the relative sky radiance. I recommend this study for publication in AMT after some minor revisions.

On section 2, it is not specified the origin of the spectral response of the camera filters shown in Fig. 1. Is it from the data sheet from the CCD manufacturer? Are they calculated somehow? Please specify. The exact setup of the RGB triband filter is not clear. How are these filters coupled to the camera body or CMOS sensor? In addition, the same as before, where does the spectral response of the filters come from?

AC: The spectral response of each RGB channel was obtained from data sheet of CMOS manufacturer, while triband filter response was provided by the sky camera manufacturer. The manuscript has been changed in order to add this information:

"*The spectral response of these filters, obtained from the data sheet from the CMOS manufacturer, is shown in Fig. 1a. An additional RGB triband filter (Fig. 1b; spectral response provided by the manufacturer) is over the full mosaic in the SONA202-NF in order to reduce the width of the colour filters*"

Regarding the setup of triband filter, it is coupled to the camera body, it is not part of the sensor, but its position is in front of the CMOS sensor.

RC: On section 3.2, what I'm understanding here is that the camera provides the images with a white balance gains of 1.1 and 2.1 for the G and B channels respectively. The white balance is reversed by dividing the pixel values by these gains (as in Eq. 1) and the everything is calculated with the image without the white balance, correct? If that is the case, the effect of having saturated pixels due to the white balance still exist. Is it possible to manually set the white balance off (or white balance gains to 1) in the camera?

AC: Yes, it is correct. This is the case, the pixels still are saturated by the white balance problem. The used camera was a prototype and it had the white balance fixed. The white balance can be set off now, but the option was not implemented when the images used in this paper were captured.

RC: On section 3.3 (paragraph starting on line 173), the M_DFS and SIGMA_DFS are average and standard deviation of the sum of the three channels? Or is it one channel (which one?)

AC: It is for all channels together (the sum of the three) since the readout noise (or dark signal) must be not dependent on the colour filter when there is not light incoming to the sensor.

"*All the dark frames have been corrected by Eq. (1), and the mean ($M_{DFS}$; mean dark frame signal) and standard deviation ($\sigma_{DFS}$) of the signal of all pixels (including the three channels) has been calculated for each dark frame.*"

RC: On the same section, finally, a temperature correction of the dark signal is not applied, right?

AC: Exactly, a temperature correction is not applied. We assume the main signal recorded under dark conditions is the black level, which is corrected. In fact, the mean of dark signal is below the pixel signal resolution of 1 DC. Hot pixels present a higher dependence on temperature, but they are also removed.

RC: On section 3.4, why the cloudy day (18th August) is used instead of a clear day (the 17th) to show the signal value at one exposure time vs other exposure times? Why adding an Fig. A1 when that could the Fig. 6? In addition, on Fig. A1, the description says 18th august and should be 17th.

AC: The original idea was to show two different cases, one full clear day and other with cloud presence, to observe that the results are similar and, hence, we can use all available days for the slope $t_j/t_i$ calculation. Unfortunately, just one figure for one day is too big and hence we decided to attach one of them as supplementary. We think the importance of both figures is equal, and it is true that for the days with clouds the figure looks worse due to the dispersed data, but these data are very low frequent because the colour density scale is logarithmic. We were not sure what figure should be as supplementary and what as Fig. 6, but finally we decided to put the clear day figure as supplementary since it needs less explanation (it does not present high deviation data).

The description of the mentioned figure has been changed.

RC: On page 8, line 233, after readout noise it should say "(N_r)" just for clarification.

AC: Yes, it is true, but N_r is defined before at the end of the Section 3.3, and hence the definition of N_r again could be redundant.

RC: On section 4.1 it is introduced that the time window for comparison is 10 minutes. Having almost 2 years of data, it might be possible to narrow this window. This could probably have an impact on the deviation of radiance values at small scattering angles (<10) besides other possible effects. As shown on Fig. 9, the slope of the radiance for small scattering angles is very steep, and a difference in sun position between image and photometer might have an impact. At least it could be quantified.

AC: We found that 99.7% of the camera-photometer data pairs chosen with the time window of 10 min were within ±2.5 min, which was expected since HDR images are recorded every 5 minutes (0.3% of the cases with more than 2.5 min are due to some interruption in camera capture). We have changed the time window threshold to ±2.5 min instead of ±10 minutes in the new version of manuscript. The results are similar to the obtained in the previous version.

RC: As a final comment, it if very surprising that the effect of the reflection of the lens on the dome is not higher. It is clearly visible on the images and, even though the

almucantar and hybrid configurations explore very specific angles, I would expect a higher impact, especially for the hybrid configuration where the angles measure pass transversally the reflection (as shown on Fig. 9).

AC: We think the effect of the reflection on the dome is unfortunately high, especially at longer wavelengths since the sky signal is lower. It is true, the effect has more impact on the hybrid scan, but the effect is high since it can be even observed in the radiance plot of Figure 9 (bottom panel), where a jump in radiance can be observed from 40º to 45º scattering angle because the radiance at 45º is inside the problematic area (reflection on the dome). The comparison between camera radiances and simulated ones revealed that the reflected area affects the radiance, and hence it is not recommended to use the sky radiance in this area (zenith angles from 48º to 65º).

---

## Author Comment (AC2) · 14 Jan 2021

**Response to the Referee #2 comments for the manuscript "Relative sky radiance from multi-exposure all-sky camera images" By Juan Carlos Antuña Sánchez et al. in AMTD**

First, we are grateful for the effort of Referee #2 and her/his review in detail. Reviewer comments are in black font (RC), and author comments (AC) in red font.

**Author's answer to Anonymous Referee #2**

RC:  General Comments
The manuscript describes the characterization of an all-sky camera to extract relative sky radiances which are then compared to measured (AERONET) and modelled radiances. Although no real new technique is revealed in the manuscript, the topic is relevant for atmospheric science and fits the scope of the journal. I would suggest a number of improvements, which add up to a major revision, to raise the overall quality and enhance the readability.

Specific Comments
RC: L11pp. Please consider to improve the description of the relative / normalized radiance - see also the respective comment below
AC: Trying to improve the description of relative radiances the abstract has been rewritten as:
"*A methodology is proposed to obtain a linear high dynamic range (HDR) image and its uncertainty, which represents the relative sky radiance (in arbitrary units) maps at three effective wavelengths. The relative sky radiances are extracted from these maps and normalized dividing every radiance of one channel by the sum of all radiances at this channel. Then, the normalized radiances are compared with the sky radiance measured at different sky points by a sun/sky photometer belonging to the Aerosol Robotic Network (AERONET).*"

RC: L14. "in line" is a little loose phrasing for a scientific abstract.
AC: We change "in line" by "correlate" in the new version:
"*The camera radiances correlate with photometer ones*"

RC: L30pp. Please rewrite the first two paragraphs in a more precise way and better English language. For example, … responsible for the colour of the blue sky under clear conditions, and also by aerosols and surface albedo (rather than clouds, in clear conditions, I guess).
AC: Both paragraphs have been modified to be clearer:
"*The knowledge of sky radiance is a fundamental problem of the radiative transfer in the atmosphere, or other media, where absorption, emission and scattering processes occur (Coulson, 1988). Restricting to the case of solar radiation in the atmosphere-surface of the Earth, sky radiance depends on the Sun position in the sky and its angular distribution is mainly controlled by the light scattering caused by atmospheric gases through the Rayleigh scattering (responsible for the colour of the blue sky under clear conditions) but also caused by aerosols and clouds through Mie scattering. The knowledge of the sky radiance is useful, among other fields, in photovoltaic production, to calculate what solar radiation reaches an oriented panel (Li and Lam, 2007) and in human health to know the*"

*solar UV radiation dose received by a human body (Seckmeyer et al., 2013; Schrempf et al., 2017).*

*The spectral sky radiance reaching the Earth surface under cloud-free conditions is basically the solar irradiance scattered by gases and aerosols, therefore the knowledge of the spectral sky radiance at different angles is a footprint of the aerosol properties; it implies the sky radiance contains useful information that can be used for the retrieval of aerosol optical and microphysical properties (Nakajima et al., 1996; Dubovik and King, 2000). In fact, even relative sky radiance measurements (in arbitrary units) are useful for this purpose (Román et al, 2017a). Most of remote sensing techniques, mainly those used by satellite platforms, are also based on upward sky radiance measurements, formed by the radiation reflected by Earth surface and scattered by atmosphere, allowing to determine the different atmospheric compounds.*"

RC: L53. "to derive sky radiance measurements..." It would be good for the introduction to state what is new/improved in the current work in relation to that past work.

AC: The text has been modified by "to derive sky radiance and luminance measurements…" since the paper of Tohsing et al. (2013) calculates luminance instead of radiance. In order to introduce what is new/improved in this work, a new sentence has been added after the description of the main objectives.

"*The novelty of this work with respect to other works that also retrieve the sky radiance with sky cameras, such as Román et al. (2012), lies among other issues in the use of raw and multi-exposure images captured with narrower spectral filters.*"

RC: L92. Are the sky radiance available at the AERONET website? If yes, where exactly?

AC: Yes, it is available at the AERONET website. The manuscript includes the sentence "*These data have been directly obtained from AERONET website (Aerosol inversions v3 - Download Tool), level 1.5 version 3 data*". The link to AERONET main webpage has been added in the Introduction section.

RC: L105. How many pixels are the threshold here?

AC: There is not a threshold in the number of pixels. The chosen exposure times are always the same for daytime mode, and hence the number of pixels with enough pixel signal without saturation changes with each sky scene.

RC: L137pp. If white balance is about the relative scaling of the channels, how does it affect the individual relative radiances, which is the subject here? If the answer is "it does not", please justify the inclusion of this section/topic.

AC: If white balance is not corrected then it does affect to the individual relative radiances since the black level and readout noise will change. The determination of black level could not be possible if the white balance is not corrected, since we will find a different black level for each channel (see Eq (1)).

There is also an important reason to include the discussion about white balance in the manuscript. As it has been observed, the direct application of a white balance factors in the raw images leads to an undesirable and unneeded pixel saturation which reduces the dynamic range of a single image. This result allows us to make a recommendation to other researchers but especially to the manufacturers: for advanced use of sky cameras, white balance factors should not be applied at least until the pure raw image is captured and saved.

RC: L152pp. Please clarify this section. In general, the measured signal is a (non-) linear function of the number of absorbed photons plus a dark signal offset plus a noise, right? So dark signal (offset) and noise should be distinguished.

AC: Yes, it is. But the noise can be divided in two: shot noise (inherent to the light) and readout noise (which is an electronic gaussian noise and it is always present even without light). In the text the dark signal (offset) is called "black level" and the noise is called "readout noise". For clarification, the next sentence has been added:

"*Summarizing, the recorded signal in one pixel is the signal generated by the number of absorbed photons plus the black level signal (offset) plus the total noise (being the readout noise part of the total noise).*".

RC: L175pp. Is there any explanation for the effect of the decrease again for temperatures above 50?

AC: We tried to look for an explanation or a similar behaviour in literature, but we did not find any relevant results. As a first explanation we thought about an hysteresis problem between the temperature measurement and the real temperature sensor. However, the temporal evolution of the recorded values and the temperature ones (see Fig. R1) does not indicate a clear hysteresis pattern.

[Figure]

Figure R1: Time series of the mean (upper panel) and standard deviation (bottom panel) of the dark frame signal as a function of sensor temperature for different exposure times and excluding hot pixels. Red crosses represent the sensor temperature (right axis).

The values of the mean and standard deviation of the dark frames are low values (below the signal resolution of 1 DC). Anyway, we assumed the readout noise for each image the maximum obtained in this experiment in order to be conservative.

Finally, the explanation of this observed behaviour is out of the objectives of this paper, which, regarding the readout noise (dark noise), only wants to report the obtained results in the experiment carried out.

RC: L179pp. The benefits of Figure 4 and 5 (left) are questionable. I am afraid, I can only see black squares.

AC: These figures lost resolution when the manuscript was converted from ".doc" to "pdf". It has been solved in the new manuscript, providing these figures separately with

high resolution. The hot pixels can be well appreciated now if the reader makes zoom in. Right panel of Figure 4 is mainly black since it is with hot pixels blocked. We thought both figures are needed since they show how the pattern about hot pixels are distributed by the CMOS pixels and how their signal highly correlates with temperature.

RC: L202pp. Is there any less confusing way of presenting these numbers for the various cases?

AC: We thought to provide this information in a figure (see Fig. R2), but we considered that it is not too much information to justify one more figure. Hence, we think the best way is to include the observed numbers in the text. We do not think the presentation is confusing, we provide 3 percentage values (one for the pixels with 0 DC, one for the pixels with 1 DC and one for the pixels with -1DC), which is straightforward. Then, in the parenthesis we divided these percentages by channel providing three more values, but they are in the parenthesis, thus making it easier for the reader.

[Figure]

Figure R2: Frequency of DFS for all pixel signals (excluding hot pixels) of all measured dark frames at all exposure times.

RC: L216pp. I find this explanation and the corresponding figure 6 quite confusing to read. Please try to improve here. Also avoid the "jet" or rainbow type colormap, as it is known to e.g. not be perceptually uniform. Maybe use the colormap of Fig. 8 throughout.

AC: The colormap has been replaced by the same of Figure 8 as the reviewer suggested. The sentence has been modified trying to be clearer, as follows:

"*Therefore, we have represented each corrected pixel signal ($PS_c$) obtained at a given exposure time as a function of the same signal but captured under other exposure time (same pixel and scenario, but different exposure times and hence different signals); it has been done for all combinations of two different exposure times.*"

RC: L301pp. Maybe clarify the nomenclature of "relative" radiance (relative to what?). Would it be more accurate to call it "corrected signal"? Later, the radiances are normalized relative to the sum of radiances.

AC: A new sentence has been added to clarify the term "relative radiance":

"*The term "relative radiance" in this work refers to uncalibrated sky radiance measurements, i.e., it is the sky radiance but in arbitrary units instead of $Wm^{-2}sr^{-1}$ or physically equivalent units.*"

RC: L329. The concept of "dispersion" is usually applied differently.
AC: We have changed the sentence by:
"*However, there are several data pairs showing higher camera radiances than photometer ones*."

RC: L334pp. Is the word "linearity" used in two different ways in the same sentence?
AC: Yes. This sentence has been replaced by:
"*The worse performance of the lowest scattering angles is not caused by problems in the linearity of HDR image, especially for high signal values, because higher scattering angles show also high radiance values, but these ones fit well with the photometer measurements*".

RC: L369pp. This whole paragraph is not really relevant for the paper (validation of the model?), nor is the conclusion surprising (r=1 for the comparison).
AC: This radiative transfer model is used to study the performance of the proposed camera sky radiances. The best way to study this performance should be the use of alternative sky radiance measurements, but these are not available at the camera effective wavelengths and, hence, we need to use a model instead of real measurements. Therefore, we consider that is necessary to know the performance of the model because if the model is not accurate it will not be useful to our purpose. It is true that the obtained result is that the model is accurate, but not to demonstrate it previously would be unwary on our part because we need to be sure that the model reproduces well the radiance measurements. These reasons motivated the inclusion of this paragraph in the manuscript.

RC: L383. "when the radiances are also normalized." It seemed that the radiances should always be normalized for all comparisons.
AC: It is true that we are mainly interested in the performance of the model to estimate normalized radiance, but we think the addition of the performance of the model to reproduce the absolute sky radiance is also useful to complement the paper. This result is interesting because in the future we could use the simulations of the model for example to calibrate in absolute values (physical units) the sky radiance values from the camera.

RC: L388. Would the wavelength difference also affect the standard deviation?
AC: Yes because the relationship between the radiance at two different wavelengths is not constant. For example, the ratio between the radiance at 467 and 440 nm depends on the aerosol and sky conditions. If this ratio would always present the same value (not changes on aerosol and sky conditions), then the wavelength difference between photometer and camera should only affect to the bias and not to the standard deviation; however, the aerosol and sky conditions frequently vary, changing the spectral variation of sky radiance, and it affects to the standard deviation obtained in the sky radiance comparison.

RC: L423pp. Please rewrite this sentence in a more comprehensible way.
AC: The sentence has been rewritten as follows:
"*In this case, camera radiances overestimate the simulated ones at the lowest angles and underestimate the simulations for the rest of angles; this fact leads to the observed increase (in absolute value) in the mean camera-simulations differences.*"

RC: L575. Fig. 2. The letter annotation in the panels is missing.
AC: Letter annotation has been added in the new manuscript.

RC: L645. Fig. 9. There are very noticeable concentric rings in the images (same in Fig. 2), the jumps are even visible in the lower radiance plot (around +/- 40 scattering angle). This looks like a major issue for the topic at hand and should be discussed accordingly. While all-sky images typically show some kind of internal reflections and artifacts, these look quite a bit worse than in many examples I have seen from other cameras.
AC: Two concentric and thin rings can be observed for the lowest zenith angles. We observed this kind of reflection in a lot of other sky camera models without shadowband (or similar) for the Sun. It is true that in the used camera (a prototype), a wider ring can be observed for the zenith angles between 48º and 65º, but we discuss about that in the manuscript. This ring is a reflection caused by the discoloration of a part of the camera, that lost part of its black colour. As it is mentioned in the paper, the camera is a prototype, and the problematic piece can be replaced by other. We are honest with the obtained results and we have no problem in showing this wider ring in the paper. In fact, we recommend not to use the pixels inside this ring, because we observed a worse agreement between camera and simulated radiances in this area. Our objective was not to obtain a sky image that "looks" better than other cameras, we aimed at obtaining and validating the spectral relative (uncalibrated) sky radiance in the most accurate way we could; hence we discard the use of the pixels inside the mentioned ring.

We modified the text to clarify that:
"*These differences appear for points with zenith angles from 48º to 65º, which corresponds with the position of the observed ring image reflected on the dome (see Figs. 2 and 9); this is an image of a piece of the prototype camera that lost part of its black colour*"

RC: Technical corrections:
I acknowledge that the authors are not native English speakers, but please make an effort to improve the language in general, in terms of grammar (even spelling) and clarity. Some examples below:
L12 maps.
L14. Except for.
L42. "Earth's" or "the earth", "the atmosphere".
L86. "measure" sky radiances.
L136. "a wider".
L138. "affects the".
L140. "multiplication of the recorded"
L306. "radius of 3"
L286. "as can be seen in Fig. 2."
Fig.12 x-labels: scattering angle. (same in Fig. 14)
AC: These technical corrections have been corrected. Moreover, the English has been revised and corrected in the new manuscript version.